# PodEval: A Multimodal Evaluation Framework for Podcast Audio Generation

## Abstract

Recently, an increasing number of multimodal (text and audio) benchmarks have emerged, primarily focusing on evaluating models' understanding capability. However, exploration into assessing generative capabilities remains limited, especially for open-ended long-form content generation. Significant challenges lie in no reference standard answer, no unified evaluation metrics and uncontrollable human judgments. In this work, we take podcast-like audio generation as a starting point and propose PodEval, a comprehensive and well-designed open-source evaluation framework. In this framework: 1) We construct a real-world podcast dataset spanning diverse topics, serving as a reference for human-level creative quality. 2) We introduce a multimodal evaluation strategy and decompose the complex task into three dimensions: text, speech and audio, with different evaluation emphasis on "Content" and "Format". 3) For each modality, we design corresponding evaluation methods, involving both objective metrics and subjective listening test. We leverage representative podcast generation systems (including open-source, close-source, and human-made) in our experiments. The results offer in-depth analysis and insights into podcast generation, demonstrating the effectiveness of PodEval in evaluating open-ended long-form audio. This project is open-source to facilitate public use: `https://anonymous.4open.science/r/PodEval-iclr`.

## 1 Introduction

With the rapid development of AIGC (AI-Generated Content) in recent years, many innovative applications have emerged. AI Podcast represents a key application scenario for audio-based generative models (Google, 2023; ByteDance, 2025). However, evaluating podcast-like audio is challenging due to: 1) it is an open-ended task, which means there is no reference standard answer; 2) the evaluation of long-form speech/audio is particularly difficult, as longer formats introduce more variability. Objective metrics often fail to capture human perceptions accurately, while subjective listening tests face issues like user inattention, which reduces the validity of results; and 3) podcasts often incorporate additional elements, like music and sound effects, making the evaluation more complicated.

To address these challenges and establish a clear evaluation framework, we decompose podcast-like audio into three dimensions: **text** (conversation transcripts), **speech** (spoken dialogue), and **audio** (speech, music, sound effects, and their interaction). While these dimensions inherently overlap, they offers a structured framework for evaluation focus. Specifically, the conversation transcripts in podcasts are primarily used for **content** (the message being conveyed) evaluation, whereas speech, music and sound effects primarily contribute to **format** (how the message is presented) evaluation.

Different modalities have their own commonly used evaluation methods. For text, metrics such as BLEU (Papineni et al., 2002), ROUGE (Lin, 2004), and METEOR (Banerjee & Lavie, 2005) focus on fluency and relevance, while newer approaches like BERTScore (Zhang et al., 2019) utilize pre-trained language models to capture semantic alignment. For speech, objective metrics like Mel Cepstral Distortion (MCD) and Perceptual Evaluation of Speech Quality (PESQ) (Rix et al., 2001) are widely used, alongside subjective evaluations like Mean Opinion Score (MOS) (Sector, 1996). For audio, metrics like Frechet Audio Distance (FAD) (Kilgour et al., 2018) and Kullback-Leibler Divergence (KL) are employed to evaluate audio quality, while listener surveys provide subjective insights. However, these evaluation methods are not directly applicable to podcast evaluation since:

- Most content-related objective metrics rely on reference scripts to measure quality and relevance. However, podcast generation lacks standardized references as it is an open-ended generation task. Moreover, relying on such references limits the diversity and creativity of the generated content.

- General speech evaluation focuses on individual sentences, while podcasts require natural and interactive dialogue, emphasizing dialogue-level naturalness. Additionally, voice presentation in multi-speaker scenarios is critical to ensuring role distinction and overall listener engagement.
- While music and sound effects are not essential to every podcast, their evaluation, when present, should go beyond the quality of individual audio events. Instead, it should focus on their overall harmony and seamless integration with the speech content to enhance the listener's experience.
- Subjective tests are essential for open-ended generative tasks, but crowdsourced data often faces reliability issues, as it is difficult to control or determine whether users are attentive. Especially for long-form content, users may lose focus or respond randomly, which affects the result validity.

In this work, we introduce **PodEval**, a comprehensive multimodal evaluation framework designed for podcast-like long-form audio generation. The contributions can be summarized as:

- We construct a real-world podcast dataset spanning a wide range of podcast categories and topics, serving as a reference for human-level creative quality. Model-based samples are also provided.
- We decompose podcast-like audio evaluation from a multimodal viewpoint—text, speech, and audio—to establish a clear evaluation framework, with distinct focuses on "Content" and "Format".
- For each modality, we design tailored metrics to address diversity considerations. For text, we combine quantitative metrics with LLM-based evaluations to assess conversation scripts. For speech and audio, we design objective metrics and subjective listening tests to evaluate spoken dialogue and overall audio performance. All evaluation methods are organized into open-source tools for ease of use. Subjective tests are enhanced by spammer detection to improve data validity.
- We utilize representative podcast generation systems in our experiments, including open-source, closed-source, and human-made ones. The results offer detailed analyses of these systems, provide insights for podcast generation, and validate the effectiveness of our evaluation framework.

## 2 RELATED WORK

### 2.1 PODCAST GENERATION

Podcasts are a popular audio format, with platforms like Apple Podcasts and Spotify leading the way. The rise of the AI podcast began with Google's NotebookLM (Google, 2023), which gained popularity in late 2024 for its "Audio Overviews" feature. This feature converts materials into conversational, two-person podcasts, praised for its highly natural dialogue speech. Similarly, most open-source podcast generation systems focus on dialogue speech synthesis, like Dia (Nari Labs, 2025), Muyan-TTS (Li et al., 2025), MoonCast (Ju et al., 2025) and MOSS-TTSD (OpenMOSS Team, 2025). These systems function primarily as dialogue Text-to-Speech (TTS) engines for text-given scenarios. Another type of podcast generation system takes a more holistic approach, incorporating elements beyond speech, such as text and music/sound. For example, WavJourney (Liu et al., 2023) leverages LLMs to connect components like TTS and Text-to-Audio (TTA), generating element-rich audio programs. Upon this, PodAgent (Xiao et al., 2025) introduces a "Host-Guest-Writer" multi-agent system to create informative conversation scripts and builds a voice pool for appropriate voice selection. Table 1 compares the systems leveraged in subsequent experiments.

Table 1: Comparison of podcast generation systems.

| System | Open-Source? | # Speaker | Support Voice Selection? | Is Dialogue TTS? | Support Music/Sound? |
|---|---|---|---|---|---|
| NotebookLM | ✗ | 2 | ✗ | - | ✗ |
| Dia | ✓ | 2 | Preset | ✓ | ✗ |
| Muyan-TTS | ✓ | 1 | Preset | ✗ | ✗ |
| MoonCast | ✓ | 2 | Preset | ✓ | ✗ |
| MOSS-TTSD | ✓ | 2 | Preset | ✓ | ✗ |
| PodAgent[*] | ✓ | N | Auto | ✗ | ✓ |

[*] PodAgent uses CosyVoice2(Du et al., 2024) as its backend TTS model, which is a single-sentence TTS system.

### 2.2 EVALUATION ON GENERATIVE MODELS

Various evaluation works have emerged along with the development of LLMs and multimodal generative models. **Text-related Evaluation**, such as SuperGLUE, MMLU, and BIG-bench (Wang et al., 2019; Hendrycks et al., 2020; Srivastava et al., 2022), assesses the capabilities of LLMs across diverse tasks with preset ground truth. Subsequently, MT-Bench (Zheng et al., 2023) explores the potential of LLMs as evaluators, and Chatbot Arena (Chiang et al., 2024) provides an open platform for

assessing LLMs based on human preferences. **Speech-related Evaluation**, such as SUPERB (Yang et al., 2021), is designed for discriminative tasks like speech recognition and speaker identification. However, evaluations for generative tasks are scarce due to their inherent diversity and subjectivity, making subjective evaluation essential for speech generation tasks. For instance, VOCBENCH (AlBadawy et al., 2022) incorporates both subjective and objective evaluations to assess vocoder performance in speech synthesis. Similarly, numerous **Audio-related Evaluation** work, such as AIR-Bench, Audiobench, MMAU, and MMAR (Yang et al., 2024; Wang et al., 2024; Sakshi et al., 2024; Ma et al., 2025), focus on audio understanding and reasoning. Subjective evaluation remains crucial for assessing audio generation systems and is typically tailored to specific generation tasks. Unlike existing evaluation works, **PodEval** introduces a comprehensive framework specifically designed for podcast-like audio generation. It emphasizes both subjective and objective evaluations across text, speech, and audio, with all metrics closely aligned with real-world user experience.

## 3 REAL-POD: REAL-WORLD PODCAST DATASET

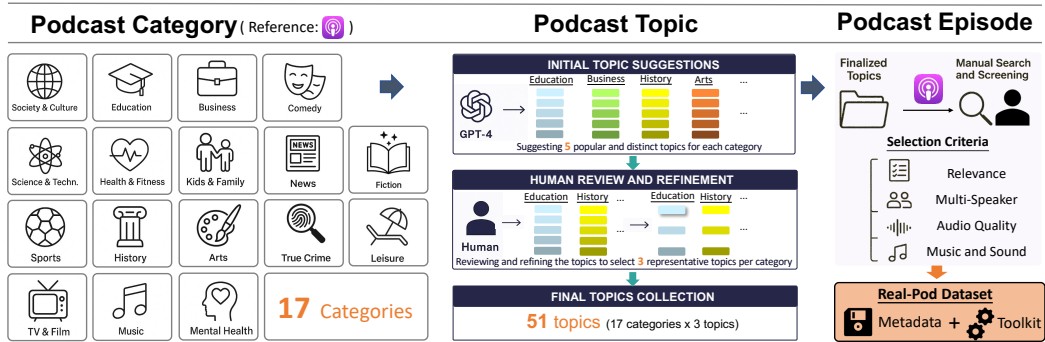

Figure 1: The workflow for constructing the Real-Pod dataset.

There is no unified standard for defining "what makes a good podcast episode." Unlike textbooks or official TV programs, podcasts can be created by anyone to share their unique ideas or insights. We do not make direct comparisons between generated podcasts and real podcasts—such comparisons are inherently unfeasible, especially when they approach topics from entirely different perspectives. Instead, we construct a real-world podcast dataset, called **Real-Pod** dataset, to serve as a reference for human-level creative quality. It is important to note that this dataset acts as a "reference" rather than an absolute "answer". The design principles of the Real-Pod dataset are **real** (consists of human-made podcasts), **broad** (diverse topic coverage) and **rich** (varied formats, like multi-speaker, music and sound). The workflow for constructing the Real-Pod dataset is illustrated in Figure 1:

- **Podcast Category**. We began by compiling a comprehensive list of podcast categories based on the taxonomy from Apple Podcast (Apple Inc.). The 17 categories are shown in Figure 1-left.

- **Podcast Topic**. Next, we established relevant topics for each category through a two-step process: (1) using GPT-4 (Achiam et al., 2023) to generate 5 popular and distinct topics per category, reflecting current trends and listener interests; and (2) manually reviewing and refining these topics to ensure their uniqueness and relevance with real-world podcasts, selecting 3 representative topics for each category, resulting in a final collection of 51 topics (17 categories × 3 topics).

- **Podcast Episode**. After finalizing the topic collection, we manually searched and screened podcast episodes to identify those most relevant to the selected topics. The selection process was guided by: (1) Topic Relevance: Episodes were selected based on their alignment with the predefined topics. (2) Rich Format: Preference was given to episodes that featured multi-speaker conversations, included background music and sound effects, and exhibited high audio quality.

## 4 TEXT-BASED EVALUATION

The dialogue content in podcasts is extracted in text format for evaluation, representing the core message the podcast aims to convey. Podcast dialogues often center around specific topics, showcasing participants' unique perspectives and insights, which makes reference-correlation-based methods infeasible. Instead, the richness of perspectives conveyed (to provide informative takeaways for the

listener) and the presentation style of the dialogue (to enhance listener comprehension) should be the primary focus of evaluation. Therefore, we follow the dialogue script-based evaluation methods proposed in PodAgent (Xiao et al., 2025), which adopt a two-fold approach: (1) **Quantitative Metrics** such as Distinct-N, Semantic-Div, MATTR, and Info-Dens to assess lexical diversity, semantic richness, vocabulary richness, and information density, respectively. These metrics operate independently of reference texts and focus on intrinsic text characteristics; (2) **LLM-as-a-Judge**, leveraging GPT-4 to replace human evaluators for complex and comprehensive assessments. Evaluation criteria include coherence, engagingness, diversity, informativeness and speaker diversity. It incorporates comparative evaluations to reduce bias and evidence-based scoring for robust and reliable results.

## 5    SPEECH-BASED EVALUATION

Speech is the core component of a podcast, serving as the medium for content delivery, and how the message delivered plays a crucial role in shaping the listening experience. To ensure a multidimensional evaluation, we first integrate the following **Objective Metrics**:

- **WER** (Word Error Rate) measures pronunciation accuracy, a critical indicator of the robustness of TTS-based podcast generation systems, powered by Whisper (Radford et al., 2022) in our toolkit.
- **DNSMOS** (Reddy et al., 2022) evaluate the speech quality (SIG), background noise quality (BAK), and overall quality (OVRL, P808_MOS) of speech. SIG, BAK, and OVRL are trained according to P.835 (ITU-T, 2003), while P808_MOS is trained based on P.808 (ITU-T, 2018).
- **SIM** stands for Speaker Similarity. In podcast generation systems, zero-shot TTS is often employed to replicate the voice of a preset speaker. The SIM between the synthesized voice and the reference voice serves as a crucial metric about vocal fidelity. In PodEval, SIM is quantified using the cosine similarity of extracted speaker embeddings (Plaquet & Bredin, 2023; Bredin, 2023).
- **SPTD** is a brand new metric we proposed, standing for Speaker Timbre Difference. As audio programs, podcasts are accessible only through listening. In multi-speaker conversations, voices with greater timbre differences enhance clarity and make the information easier to follow. SPTD is to assess the overall timbre variation across speakers. Equation 1 calculates the SPTD among $N$ distinct speakers.

$$\text{SPTD} = 1 - \frac{2}{N(N-1)} \sum_{i=1}^{N} \sum_{j=i+1}^{N} \text{sim}(\mathbf{e}_i, \mathbf{e}_j) \tag{1}$$

Objective metrics can be calculated efficiently at a low cost without human involvement. However, the **Subjective Listening Test** remains a necessary indicator of human perception. Unlike general speech synthesis, which emphasizes sentence-level pronunciation accuracy and naturalness, podcast speech focuses on achieving human-like natural dialogue. Subjective tests for such long-form speech present several key **challenges**: 1) the length of dialogue in podcasts ranges from a few minutes to over an hour, making it impractical to evaluate the entire speech directly; 2) the difficulty of comparing more than two systems simultaneously; 3) guiding user focus toward dialogue naturalness, rather than on factors like content; 4) balancing topic diversity within a fixed testing capacity; and 5) ensuring that crowdsourced evaluators remain focused and provide reliable feedback.

In PodEval, we design the **Dialogue Naturalness Evaluation** based on the MUSHRA framework (Schoeffler et al., 2018). The key insight from this framework, *incorporating both high-quality and low-quality anchors*, helps evaluators establish a reliable reference of quality range. For researchers, analyzing scores for these anchors helps identify inattentive evaluators, enabling the *filtering of invalid submissions* and improving the data vadility. In our task, we use real podcast segments from the *Real-Pod dataset as the high-quality anchor* and synthesized dialogue segments from *eSpeak Contributors as the low-quality anchor*. For podcast samples from different systems, we provide an automatic toolkit to extract dialogue segments featuring *turn-taking* between speakers, representing a typical dialogue flow. Each dialogue segment is extracted with a *preset length* (e.g. 15–25 seconds) to ensure the speech samples are of similar duration. We select dialogue segments from *all 17 categories* in the Real-Pod dataset to ensure content diversity while keeping the total listening test duration *within 30 minutes*. In each test group, samples from different systems are presented *on the same page*, along with a *reference Real-Pod sample* to guide evaluators on what a natural dialogue sounds like. The scoring is adjusted using a slider ranging from 0 to 100, divided into *five stages with a clear definition*. Detailed instructions and website design can be found in Appendix A.3. [1]

---

[1] The demo website is hosted at `https://podeval.github.io/PodEval-Subjective/?config=dialogue.yaml`. Everyone is welcome to try it out and view the results at the end.

## 6  AUDIO-BASED EVALUATION

In this section, we introduce the audio-based evaluation for podcasts, which treats speech as one component and assesses the overall audio performance, including speech, music and sound effects (MSE), and their interactions. Similarly, we first introduce the following **Objective metrics**:

- **Loudness**: Loudness ensures audio falls within an acceptable volume range. The ITU-R BS.1770-4 standard (BS Series, 2011) is widely recognized for measuring audio loudness and true-peak levels. Based on this, the (EBU R128, 2011) standard has been broadly adopted by broadcast and streaming platforms, recommend a target Integrated Loudness (LOUD-IT) of -23 LUFS (±1 LUFS), True Peak (LOUD-TP) $\leq -1$ dBTP and Loudness Range (LOUD-RA) $< 20$ LU. For podcast-like streaming, adjustments are made for typical listening environments, such as mobile devices where headphones are commonly used. In these cases, the LOUD-IT is recommended as -18, -16 (±1) or -14 LUFS (AES, 2021; Apple, 2023; Spotify, 2023). Netflix recommends keeping LOUD-RA between 4 and 18 LU (Netflix, 2024). There is no "absolute right" reference for loudness metrics. We propose the following reference standards considering all above guidelines: **LOUD-IT:** $-18$ to $-14$ LUFS; **LOUD-TP:** $\leq -1$ dBTP; **LOUD-RA:** 4 to 18 LU. Based on this "relatively correct" reference, we can analyze the distribution of loudness metrics across different systems. We also provide a quantitative scoring strategy in Appendix A.4.

- **SMR** (Speech-to-Music Ratio): MSE are typically integrated into podcast audio to enhance the overall listening experience. Since speech is the primary focus in podcasts, it is essential to ensure that MSE dose not overpower or mask the speech, maintaining clarity and intelligibility of the dialogue. SMR measures the balance between speech and MSE, with a minimum requirement of being greater than 0. SMR_SCORE is the proportion of cases where SMR exceeds 0.

- **CASP** (MSE-Speech Harmony): Harmony between speech and MSE is an advanced requirement. Appropriate MSE can enhance audio engagement, while discordant MSE distracts and negatively impacts the experience. The DualScore, calculated by the CASP framework proposed in Tian et al. (2025), measures the correlation between audio and speech. In PodEval, we employ the CASP model, pretrained on ∼1,000 hours of podcast data, to assess MSE-Speech Harmony.

**Subjective Listening Test** is primarily designed based on the perceptions of real users. A key challenge lies in how to evaluate extra-long audios. As we mentioned above, podcasts in the real world range from a few minutes to over an hour in length. Conducting listening tests on full-length podcast episodes is impractical due to the time, effort, and financial resources required. Moreover, it is hard to judge podcasts of vastly different lengths in a fair and consistent manner. Research on long-form audio evaluation is limited. Clark et al. (2019) did investigation on long-form speech evaluation and found that multiple evaluations are necessary due to the low correlation observed across different experimental settings. Cambre et al. (2020) conducted a comprehensive evaluation of voice selection for long-form content; however, the minimum required listening time was only 10 seconds. A podcast-related evaluation study (Austria, 2007) designed a questionnaire with carefully crafted questions in terms of both content and presentation to assess domain-specific podcasts. Different from that, PodEval does not constrain the domain of podcasts, and open-ended content evaluation is separately conducted in the text-based evaluation section. In this audio-based evaluation, we focus on assessing the overall performance of the audios. The design approach is as follows:

- We design it as a **MOS test**, where evaluators listen to one audio sample at a time and provide judgments based on predefined criteria. Compared to comparative methods, this approach is more suitable for long-form content by avoiding attention overload and consistency compromising.

- The test data are preprocessed by extracting **the first / middle / final minute**. These segments are concatenated into a single audio, separated by a beep signal. This method unifies podcast length, captures overall performance from diverse positions, and minimizes content-related biases.

- The judgment session consists of a **questionnaire** with 8 questions covering multiple dimensions, integrating both perceptual (e.g., "Information Delivery Effectiveness") and preference-based (e.g., "Speaker Expression Preference") questions. This distinction helps clarify whether the ratings are rooted in objective perception or subjective preference. Users are also asked about their willingness to listen to the full episode and the perceived human likelihood, offering insights into interest levels and audio naturalness. The detailed content can be found in Appendix A.5.2.

- We implement two strategies to enhance the validity of the collected data. **1) Attention-check questions:** These include questions like *Q1. How many speakers are there in the podcast?* and

*Q7. If music or sound effects... (Select Neutral if none are present).* These questions have standard answers, allowing us to determine whether users are actively listening to the audio. **2) Justification for answers:** Users have to provide justifications for their responses to each question, which can be short but are required. This requirement significantly increases users' focus and we can collect more detailed information from their justification. By employing these two strategies, data validity is enhanced by promoting attentiveness and filtering out unreliable responses.

# 7 EXPERIMENTS

## 7.1 TEXT-BASED EVALUATION

The text-based evaluation is conducted among GPT-4, PodAgent, MoonCast, and Real-Pod. Other systems in Table 1 are excluded as they do not provide conversation scripts. PodAgent, with its Host-Guest-Writer multi-agent system, can directly generate podcast scripts based on a given topic. While MoonCast functions similarly to NotebookLM, requiring external knowledge sources but providing prompt template for spontaneous script generation. For this evaluation, the MoonCast system uses the podcast scripts generated by PodAgent as input and transforms them into a spontaneous version.

**Quantitative Metrics.** Detailed scores calculated across 17 podcast categories for each system are presented in Appendix A.2.1. For a concise and clearer comparison, we present the overall performance (averaged across all 17 categories) in Figure 2, where we can observe that: 1) For each quantitative metric, PodAgent outperforms directly prompting GPT-4; 2) When comparing LLM-based methods (GPT-4, PodAgent) with human-created podcasts (Real-Pod), Real-Pod scores lower on lexical diversity (Distinct-2 and MATTR) but higher on information density and semantic diversity (Info-Dens and Sem-Div). This is reasonable for: i) real human interactions often include filler words and use simpler language; ii) most real podcasts are significantly longer (30 minutes to an hour), leading to higher information richness compared to generated podcasts, which are usually only a few minutes long; 3) As a spontaneous version of PodAgent, MoonCast shows reduced lexical diversity and information density. While its semantic diversity remains comparable to PodAgent.

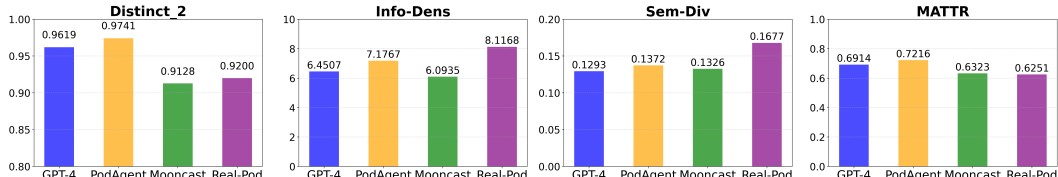

Figure 2: **Quantitative metrics:** comparison among GPT-4, PodAgent and Real-Pod.

**LLM-as-a-Judge.** This evaluation compares PodAgent (scored from -3 to 3) with GPT-4 (reference score as 0), both of which generate conversation scripts without external knowledge resources. Detailed scores for each category are provided in Appendix A.2.2. We present the overall performance and results for five specific categories in Table 2 for analysis. We can see that scores across all metrics and all categories are positive, demonstrating that PodAgent significantly outperforms directly prompting GPT-4 in generating podcast scripts across all evaluated dimensions.

Table 2: **LLM-as-a-Judge:** comparison between GPT-4 and PodAgent (overall performance and 5 specific categories). Scores range from -3 to 3, where positive values favor PodAgent.

| Metrics | Overall | Fiction | Education | Business | TrueCrime | Health & Fitness |
|---|---|---|---|---|---|---|
| Coherence | 0.7059 | 0.5000 | 0.8333 | 1.0000 | 1.0000 | 0.6667 |
| Engagingness | 1.0294 | 1.1667 | 1.0000 | 1.1667 | 0.6667 | 1.1667 |
| Diversity | 1.1765 | 1.3333 | 1.0000 | 1.3333 | 0.8333 | 1.5000 |
| Informativeness | 1.6078 | 1.5000 | 1.6667 | 2.0000 | 1.1667 | 1.6667 |
| Speaker Difference | 1.0637 | 0.9167 | 1.0000 | 1.1667 | 0.6667 | 1.0000 |
| Overall | 1.3064 | 1.2500 | 1.3333 | 1.6667 | 0.8333 | 1.2500 |

## 7.2 SPEECH-BASED EVALUATION

To ensure fairness, all open-source TTS systems use the same PodAgent-generated scripts. Subjective tests use a spontaneous version from MoonCast, while objective evaluations use the original PodAgent scripts, as filler words in the spontaneous version challenge metrics like WER.

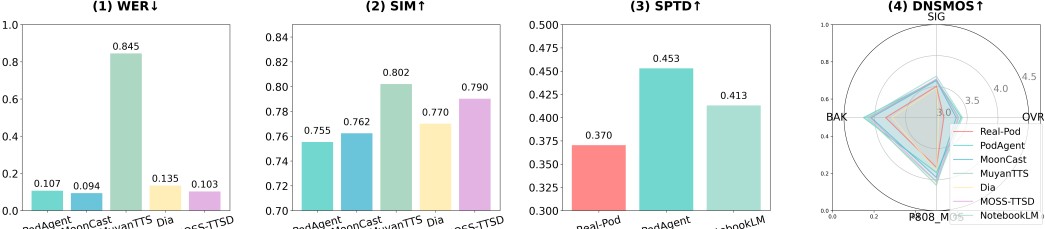

Figure 3: **Speech-based evaluation:** objective metrics (WER, SPTD, SIM, DNSMOS).

**WER.** Figure 3-(1) shows the WER results calculated for the entire conversation script. All systems, except MuyanTTS, achieve WER scores below 20%. Analysis of sampled MuyanTTS outputs reveals robustness issues like repeated sentences and the insertion of unknown content.

**SIM.** The SIM metric evaluates zero-shot TTS systems' ability to replicate the timbre of a reference voice. PodAgent, MoonCast, MuyanTTS, Dia, and MOSS-TTSD—are assessed as shown in Figure 3-(2). Each system uses the reference voice selected by PodAgent for the topic. The performance rankings are: MuyanTTS, MOSS-TTSD, Dia, MoonCast, and PodAgent. PodAgent's relatively low score in this metric likely stems from its instruction-following style control strategy. While this approach enhances overall conversational expressiveness, it can reduce speaker similarity.

**SPTD.** Figure 3-(3) shows timbre variation across speakers in the conversation for three systems: Real-Pod, PodAgent, and NotebookLM. Real-Pod reflects real-world podcasts, PodAgent uses a voice selection mechanism for distinct voices, and NotebookLM fixed voices (one male, one female). The SPTD scores rank as follows: PodAgent, NotebookLM, and Real-Pod. This likely reflects that real-world podcasts prioritize guest expertise and availability over timbre differences. PodAgent demonstrates an effective automated voice selection process for podcast creation.

**DNSMOS.** The DNSMOS metric was applied to all systems to evaluate speech quality as in Figure 3-(4). PodAgent, MoonCast, MuyanTTS, MOSS-TTSD, and NotebookLM achieve similar scores, while Real-Pod and Dia show noticeable declines in speech quality. For Real-Pod, the lower scores are due to: (1) real podcasts often use MSE for enhancement, requiring speech-MSE separation before evaluation, which may leave residual MSE artifacts, and (2) human-created podcasts involve recording, editing, or post-processing that introduce noise or instability. Dia struggles with long-form speech synthesis. Its outputs for lengthy podcast scripts frequently feature overly fast speaking speeds and occasional sentence truncations, leading to its relatively low DNSMOS performance.

Table 3: Dialogue Naturalness Evaluation - statistical information for filtering.

| Judger | 1 | 2 | 3 | 4 | 5 | 6 | 7 | 8 | 9 | 10 |
|---|---|---|---|---|---|---|---|---|---|---|
| LQ Last (%) | 94.12 | 100 | 100 | 100 | 100 | 100 | 100 | 100 | 100 | 100 |
| HQ Top2 (%) | 88.24 | 88.24 | 58.82 | 58.82 | 94.12 | 64.71 | 17.65 | 58.82 | 64.71 | 94.12 |
| **Judger** | **11** | **12** | **13** | **14** | **15** | **16** | **17** | **18** | **19** | **20** |
| LQ Last (%) | 94.12 | 100 | 100 | 100 | 100 | 100 | 100 | 76.47 | 100 | 100 |
| HQ Top2 (%) | 94.12 | 82.35 | 64.71 | 82.35 | 88.24 | 58.82 | 64.71 | 47.06 | 52.94 | 35.29 |

**Dialogue Naturalness Evaluation.** We released the task on Prolific[2], requesting 20 native English-speaking participants from the US/UK. We set the filter rules as: 1) Over 90% of LQ samples must be marked as the worst, as the synthesized samples from eSpeak are obviously robotic and unnatural. 2) Over 50% of HQ samples must rank in the top-2 best. While it is possible for other systems to

---

[2]https://www.prolific.com/

achieve a better score than the real podcast, the evaluation of the real podcast should also remain above average. Table 3 presents the two statistical metrics for the submission results. Based on these rules, Judger-7, 18 and 20 can be excluded. We also provide the box plot for each Judger in Figure 9 in the appendix for more advanced analysis. For instance, apart from the LQ samples, Judger-20 assigns similar scores to all other systems, further confirming the invalidity of this submission.

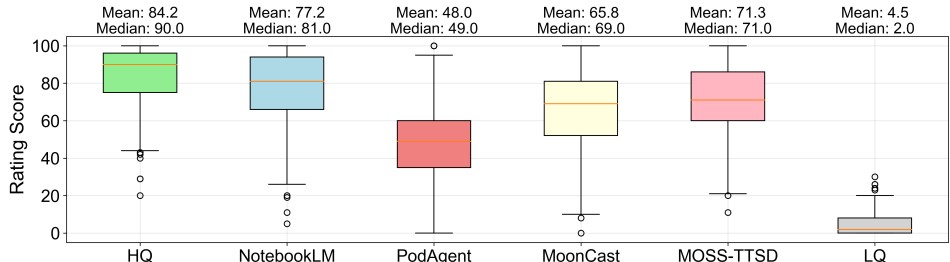

Figure 4: Dialogue Naturalness Evaluation - overall result.

*Result Analysis.* After excluding unqualified submissions, we analyzed system performance based on the remaining 17 valid submissions. Figure 4 presents the final results. We can observe that dialogue segments from real podcasts (HQ) achieved the highest scores, which aligns with expectations. NotebookLM, a closed-source product, ranked second, reflecting the high naturalness of its synthesized dialogue speech. Among the three open-source podcast generation systems, PodAgent scored the lowest, which is reasonable since its backend TTS system, CosyVoice2, is limited to single-sentence synthesis. In contrast, MoonCast and MOSS-TTSD, which support direct dialogue synthesis, performed better in dialogue naturalness evaluations. Overall, the evaluation results align with expectations, validating the rationality and effectiveness of our evaluation method design.

## 7.3 AUDIO-BASED EVALUATION

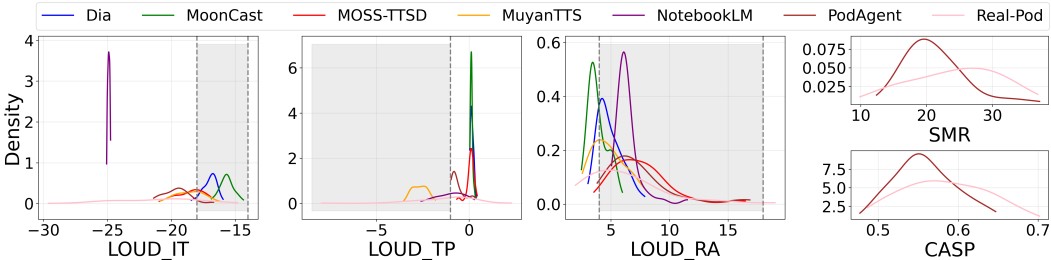

Figure 5: Density distributions of audio-based objective metrics.

**Loudness.** Figure 5 presents the density distribution of loudness-related metrics, enabling a comparative analysis with the reference range. All seven systems are included. For **LOUD_IT**, Dia and MoonCast align well with the reference range, while NotebookLM's loudness centers around -25. Real-Pod, as manually produced audio, shows a highly scattered loudness distribution. For **LOUD_TP**, Muyan-TTS performs best, with all samples maintaining a true peak loudness below -1. In contrast, MoonCast, Dia and MOSS-TTSD perform poorly, while Real-Pod continues to exhibit scattered results. For **LOUD_RA**, MoonCast has a relatively narrow loudness variation range, while PodAgent and MOSS-TTSD display richer variance. Quantitative scores are detailed in Table 9.

**SMR and CASP.** PodAgent and Real-Pod are evaluated for these MSE-related metrics. From the density distribution in Figure 5, PodAgent exhibits a more concentrated distribution compared to Real-Pod. For **SMR**, the SMR_SCORE in Table 9 shows that all PodAgent samples achieve an SMR greater than 0, whereas some Real-Pod cases fail to meet this requirement. For **CASP**, a higher score indicates better MSE-Speech harmony. Real-Pod demonstrates a higher upper limit, which is expected as exceptional human artistic creations naturally surpass AI-generated outputs. However, PodAgent delivers more consistent performance, and the overall gap between the two systems is not significant, making it an alternative way to enhance creative efficiency.

**Questionnaire-based MOS Test.** We recruited native English speakers from Prolific for this test. Section 6 describes our final test design. Prior to this, we conducted a *Pilot Test* using the questionnaire design in Figure 10. Based on feedback, we made the following improvements: 1) Reduced the scoring scale from 10 to 5 with clear definitions to reduce ambiguity and improve consistency. 2) Refined the questions to introduce perceptual and preference-based considerations. 3) Added a justification requirement for each question. These changes increased the pass rate from 75% to 90%.

In addition to direct scores, we also derive a corresponding score based on users' justifications. Specifically, given justification texts from multiple systems for the same question, GPT-4 uses the following prompt to score: *"For each system, summarize the corresponding comments into one sentence and assign a score between 1 and 5."* A detailed experiment setup is provided in Appendix A.5.2, and separate scores are listed in Table 10. Figure 6 shows the final results, averaging the direct score and the justification-based score. From the result, we can observe that:

- *Speech (naturalness and authenticity) is the most dominant factor affecting the listener's experience.* In Section 7.2, PodAgent scored low in dialogue naturalness due to using a single-sentence synthesis TTS system, leading to consistently poor results in this MOS test. This outcome is expected, as dialogue speech is the core component of podcast-like audio programs. Although PodAgent's Music/Sound (harmony) score is below Real-Pod (consistent with the results of objective metric - CASP), it is significantly higher than its scores in other metrics, indicating that *the gap between PodAgent and Real-Pod in music harmony is smaller than in speech naturalness.*

- *Real podcasts perform best in most metrics (5/7).* Real-Pod significantly outperforms other systems on holistic metrics like Engagement Level (EL) and Human Likelihood (HL). However, Full Episode Willingness (FEW) scores are low across all systems, with NotebookLM and Real-Pod scoring similarly. *This highlights the value of perceptual and preference-based question design in the test.* FEW, a preference-based question, garnered justifications like "the topic is not of interest to me" for lower scores. In contrast, higher scores for EL and HL indicate that users tend to exclude subjective factors (e.g., personal topic interest) when rating audio performance. A similar pattern is observed in Information Delivery (effectiveness) and Speaker Expression (preference).

- In the Audio Quality metric, while PodAgent and MOSS-TTSD score lower than Real-Pod, PodAgent performs better here than in other metrics, and NotebookLM slightly surpasses Real-Pod. As noted, human-made podcasts often exhibit inconsistent audio quality due to complex production. User feedback, like "Little bit of mic hiss/bloom but otherwise fine," supports this observation. This highlights that *when conversational realism approaches that of real speech, AI-based methods offers an advantage in their controllability and consistency in producing high-quality audio.*

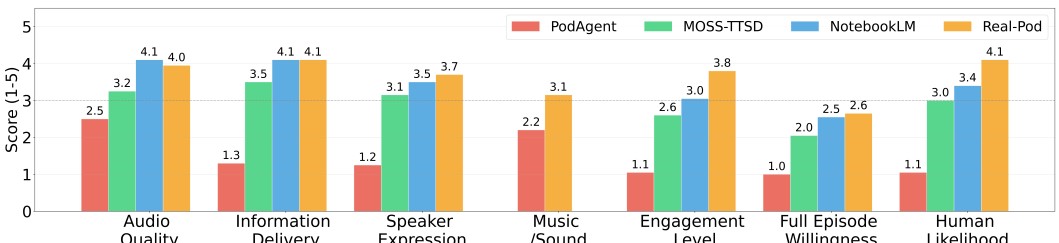

Figure 6: Questionnaire-based MOS test

## 8 CONCLUSION

PodEval is the first comprehensive evaluation framework for podcast-like audio generation, tackling the challenges of assessing open-ended, long-form content. We constructed a real-world podcast dataset as a benchmark for human-level creative quality across diverse topics and formats. By decomposing evaluation into text, speech, and audio, PodEval introduced multidimensional methods combining objective metrics and well-designed subjective listening tests. Experiments with various podcast generation systems, including open-source, closed-source, and human-made examples, validated the framework's effectiveness. The results offer insights into the strengths and weaknesses of different systems (e.g. Figure 14), highlighting PodEval's role in advancing podcast generation research and inspiring future work on evaluating open-ended, long-form content generation task.

## 9 ETHICS STATEMENT

This work introduces PodEval, a comprehensive framework for evaluating podcast-like audio generation, with careful consideration of ethical implications. The *Real-Pod dataset* was constructed using publicly available podcasts in alignment with fair use, avoiding sensitive or private data. Instead of directly providing audio files, the dataset offers publicly accessible download links and download toolkit to reduce the risk of misuse and ensure proper attribution. *Subjective evaluations* were conducted using crowdsourced workers recruited through the Prolific platform, with compensation exceeding the platform's minimum wage requirements. Reliability was ensured through attention-check questions and clear instructions for participants. To mitigate bias, the framework incorporates *diverse topics and evaluators*, promoting inclusivity and fairness. While PodEval aims to advance AI-assisted podcast generation, we emphasize its role as a tool to enhance, not replace, human creativity. PodEval is designed to foster innovation while adhering to principles of transparency, fairness, and ethical AI development.

## 10 REPRODUCIBILITY STATEMENT

To ensure the reproducibility of our work, the PodEval framework is fully open-source and accessible at `https://anonymous.4open.science/r/PodEval-iclr` (an anonymized version to comply with conference requirements). The repository contains all necessary datasets, scripts, and tools to replicate the experiments described in this paper.

### HOW TO USE THE REPOSITORY

1. Clone the Repository.
2. Set Up the Environment according to the README.
3. Proccess dataset or Run Evaluations following the corresponding instructions.

### REPOSITORY STRUCTURE

The repository is organized into the following directories:

- **Real_Pod/**
  - Provides the *Real-Pod dataset*, a curated collection of real-world podcast episodes. Includes 51 topics across 17 categories, representing diverse podcast scenarios.
  - See `Real_Pod/README.md` for dataset preparation and usage instructions.
- **Text_Eval/**
  - Tools for *text-based evaluation* of dialogue scripts. Includes both *Quantitative Metrics* and *LLM-as-a-Judge* methods.
  - See `Text_Eval/README.md` for instructions on running text evaluations.
- **Speech_Audio_Objective_Evaluation/**
  - Toolkit for *objective evaluation* of podcast audio and speech quality. Includes DNSMOS, WER, SIM, SPTD, Loudness, SMR, and CASP.
  - See `Speech_Audio_Obj_Eval/README.md` for metric calculations and usage.
- **Subjective_Listening_Tests/**
  - Framework for *subjective human evaluations* of podcast speech and audio. One is *Dialogue Naturalness Evaluation* and the other one is *Questionnaire-based MOS Test*.
  - See `Subjective_Listening_Tests/README.md` for test setup and implementation details. We also provide *website demo* link1 link2 that allow users to intuitively view the test design and participate it.

By following the provided instructions and leveraging the structured tools within each directory, users can reproduce all experiments and adapt **PodEval** for further research.

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

# A APPENDIX

## A.1 USE OF LARGE LANGUAGE MODELS

Large Language Models (LLMs) utilized in this work are as follows:(1) *Topics Initiation* during data processing of the Real-Pod dataset, which is elaborated in Section 3. (2) *LLM-as-a-Judge* method in text-based evaluation, which is illustrated in Section 4. (3) *Summarized Users' Justifications* in the Questionnaire-based MOS Test, which is described in Section 7.3 (Questionnaire-based MOS Test).

## A.2 TEXT-BASED EVALUATION

### A.2.1 QUANTITATIVE METRICS

Table 4: **GPT-4**: Quantitative metrics in text-based evaluation.

| Metrics | Overall | Fiction | Education | Business | True Crime | Health & Fitness |
|---------|---------|---------|-----------|----------|------------|------------------|
| Distinct_2 | 0.9619 | 0.9643 | 0.9588 | 0.9567 | 0.9689 | 0.9638 |
| Info-Dens | 6.4507 | 6.5865 | 6.4569 | 6.3213 | 6.6541 | 6.3880 |
| Sem-Div | 0.1293 | 0.1204 | 0.1115 | 0.1214 | 0.1443 | 0.1106 |
| MATTR | 0.6914 | 0.7027 | 0.6989 | 0.6933 | 0.6831 | 0.6870 |

| Metrics | Sports | Comedy | History | News | TV & Film | Society & Culture |
|---------|--------|--------|---------|------|-----------|-------------------|
| Distinct_2 | 0.9536 | 0.9633 | 0.9471 | 0.9486 | 0.9678 | 0.9659 |
| Info-Dens | 6.4228 | 6.2256 | 6.3792 | 6.3225 | 6.7614 | 6.6473 |
| Sem-Div | 0.1248 | 0.1356 | 0.1451 | 0.1208 | 0.1553 | 0.1507 |
| MATTR | 0.6973 | 0.6922 | 0.6905 | 0.6756 | 0.6903 | 0.6901 |

| Metrics | Arts | Leisure | Music | Kids | Mental Health | Science & Tech |
|---------|------|---------|-------|------|---------------|----------------|
| Distinct_2 | 0.9675 | 0.9729 | 0.9555 | 0.9559 | 0.9699 | 0.9710 |
| Info-Dens | 6.5054 | 6.5233 | 6.4119 | 6.2310 | 6.4787 | 6.3454 |
| Sem-Div | 0.1374 | 0.1117 | 0.1320 | 0.1229 | 0.1247 | 0.1286 |
| MATTR | 0.6885 | 0.7136 | 0.6677 | 0.6884 | 0.6994 | 0.6960 |

Table 5: **PodAgent**: Quantitative metrics in text-based evaluation.

| Metrics | Overall | Fiction | Education | Business | True Crime | Health & Fitness |
|---|---|---|---|---|---|---|
| Distinct_2 | 0.9741 | 0.9743 | 0.9730 | 0.9758 | 0.9796 | 0.9825 |
| Info-Dens | 7.1767 | 7.3791 | 7.2163 | 7.1126 | 7.1810 | 7.2927 |
| Sem-Div | 0.1372 | 0.1384 | 0.1210 | 0.1254 | 0.1514 | 0.1171 |
| MATTR | 0.7216 | 0.7399 | 0.7291 | 0.7258 | 0.7263 | 0.7386 |
| Metrics | Sports | Comedy | History | News | TV & Film | Society & Culture |
| Distinct_2 | 0.9678 | 0.9808 | 0.9483 | 0.9735 | 0.9782 | 0.9744 |
| Info-Dens | 7.1239 | 7.1600 | 7.1004 | 7.1282 | 7.3311 | 6.9568 |
| Sem-Div | 0.1487 | 0.1236 | 0.1543 | 0.1379 | 0.1690 | 0.1344 |
| MATTR | 0.7183 | 0.7248 | 0.6752 | 0.7156 | 0.7274 | 0.7119 |
| Metrics | Arts | Leisure | Music | Kids | Mental Health | Science & Tech |
| Distinct_2 | 0.9701 | 0.9790 | 0.9739 | 0.9747 | 0.9815 | 0.9725 |
| Info-Dens | 7.1977 | 7.3227 | 7.0558 | 7.0930 | 7.1822 | 7.1711 |
| Sem-Div | 0.1283 | 0.1445 | 0.1440 | 0.1353 | 0.1259 | 0.1331 |
| MATTR | 0.7101 | 0.7275 | 0.7114 | 0.7279 | 0.7328 | 0.7249 |

Table 6: **MoonCast**: Quantitative metrics in text-based evaluation.

| Metrics | Overall | Fiction | Education | Business | True Crime | Health & Fitness |
|---|---|---|---|---|---|---|
| Distinct_2 | 0.9128 | 0.9219 | 0.8998 | 0.8952 | 0.9132 | 0.9478 |
| Info-Dens | 6.0935 | 6.3613 | 5.9779 | 5.9931 | 6.0388 | 6.4230 |
| Sem-Div | 0.1326 | 0.1515 | 0.1079 | 0.1326 | 0.1405 | 0.1324 |
| MATTR | 0.6323 | 0.6598 | 0.6237 | 0.6310 | 0.6391 | 0.6698 |
| Metrics | Sports | Comedy | History | News | TV & Film | Society & Culture |
| Distinct_2 | 0.9159 | 0.9232 | 0.9169 | 0.9047 | 0.9408 | 0.8959 |
| Info-Dens | 6.1933 | 6.1729 | 6.2229 | 5.9672 | 6.2855 | 5.8031 |
| Sem-Div | 0.1451 | 0.1311 | 0.1460 | 0.1176 | 0.1318 | 0.1282 |
| MATTR | 0.6402 | 0.6435 | 0.6276 | 0.6111 | 0.6595 | 0.6121 |
| Metrics | Arts | Leisure | Music | Kids | Mental Health | Science & Tech |
| Distinct_2 | 0.9252 | 0.8889 | 0.8957 | 0.9039 | 0.9222 | 0.9073 |
| Info-Dens | 6.2335 | 5.9713 | 5.9411 | 5.9291 | 6.0298 | 6.0459 |
| Sem-Div | 0.1444 | 0.1309 | 0.1227 | 0.1291 | 0.1187 | 0.1432 |
| MATTR | 0.6370 | 0.6124 | 0.6035 | 0.6183 | 0.6321 | 0.6277 |

Table 7: **Real-Pod**: Quantitative metrics in text-based evaluation.

| Metrics | Overall | Fiction | Education | Business | True Crime | Health & Fitness |
|---|---|---|---|---|---|---|
| Distinct_2 | 0.9200 | 0.9292 | 0.9275 | 0.9049 | 0.9169 | 0.9273 |
| Info-Dens | 8.1168 | 8.2849 | 8.1160 | 7.7755 | 8.5675 | 7.9301 |
| Sem-Div | 0.1677 | 0.1776 | 0.1579 | 0.1433 | 0.1906 | 0.1646 |
| MATTR | 0.6251 | 0.6313 | 0.6346 | 0.6041 | 0.6261 | 0.6380 |

| Metrics | Sports | Comedy | History | News | TV & Film | Society & Culture |
|---|---|---|---|---|---|---|
| Distinct_2 | 0.9244 | 0.8994 | 0.9272 | 0.9100 | 0.9201 | 0.8932 |
| Info-Dens | 8.0993 | 8.2755 | 8.8282 | 7.7886 | 8.4005 | 7.7375 |
| Sem-Div | 0.1919 | 0.1660 | 0.1845 | 0.1618 | 0.1784 | 0.1701 |
| MATTR | 0.6434 | 0.5999 | 0.6304 | 0.6102 | 0.6363 | 0.5823 |

| Metrics | Arts | Leisure | Music | Kids | Mental Health | Science & Tech |
|---|---|---|---|---|---|---|
| Distinct_2 | 0.9111 | 0.9242 | 0.9420 | 0.9092 | 0.9298 | 0.9439 |
| Info-Dens | 8.1093 | 7.6949 | 8.0925 | 7.7708 | 8.2119 | 8.3031 |
| Sem-Div | 0.1653 | 0.1591 | 0.1761 | 0.1492 | 0.1668 | 0.1485 |
| MATTR | 0.6063 | 0.6176 | 0.6513 | 0.6200 | 0.6373 | 0.6582 |

### A.2.2  LLM-AS-A-JUDGE

Table 8: **LLM-as-a-Judge: comparison between GPT-4 and PodAgent.** Scores range from -3 to 3. Positive values indicate that PodAgent outperforms GPT-4; Negative values suggest the opposite.

| Metrics | Overall | Fiction | Education | Business | True Crime | Health & Fitness |
|---|---|---|---|---|---|---|
| Coherence | 0.7059 | 0.5000 | 0.8333 | 1.0000 | 1.0000 | 0.6667 |
| Engagingness | 1.0294 | 1.1667 | 1.0000 | 1.1667 | 0.6667 | 1.1667 |
| Diversity | 1.1765 | 1.3333 | 1.0000 | 1.3333 | 0.8333 | 1.5000 |
| Informativeness | 1.6078 | 1.5000 | 1.6667 | 2.0000 | 1.1667 | 1.6667 |
| Speaker Difference | 1.0637 | 0.9167 | 1.0000 | 1.1667 | 0.6667 | 1.0000 |
| Overall | 1.3064 | 1.2500 | 1.3333 | 1.6667 | 0.8333 | 1.2500 |

| Metrics | Sports | Comedy | History | News | TV & Film | Society & Culture |
|---|---|---|---|---|---|---|
| Coherence | 0.5000 | 0.8333 | 1.1667 | 0.6667 | 0.8333 | 0.1667 |
| Engagingness | 1.1667 | 1.5000 | 1.5000 | 0.6667 | 0.1667 | 0.6667 |
| Diversity | 1.1667 | 1.8333 | 1.5000 | 1.3333 | 1.3333 | 0.8333 |
| Informativeness | 1.5000 | 2.1667 | 1.5000 | 2.0000 | 1.3333 | 0.8333 |
| Speaker Difference | 1.1667 | 1.5000 | 1.1667 | 1.3333 | 1.1667 | 1.3333 |
| Overall | 1.5000 | 1.8333 | 1.5000 | 1.5000 | 0.8333 | 0.5000 |

| Metrics | Arts | Leisure | Music | Kids | Mental Health | Science & Tech |
|---|---|---|---|---|---|---|
| Coherence | 0.6667 | 0.5000 | 0.6667 | 0.5000 | 0.3333 | 1.1667 |
| Engagingness | 1.1667 | 1.1667 | 1.1667 | 1.0000 | 0.8333 | 1.3333 |
| Diversity | 1.1667 | 1.1667 | 1.0000 | 1.0000 | 0.3333 | 1.3333 |
| Informativeness | 1.8333 | 2.0000 | 1.8333 | 1.3333 | 1.1667 | 1.8333 |
| Speaker Difference | 1.3333 | 1.1667 | 0.8333 | 0.8333 | 0.6667 | 0.8333 |
| Overall | 1.5000 | 1.6667 | 1.5000 | 1.1667 | 0.8333 | 1.5417 |

## A.3    Speech-based evaluation (Subjective)

**Welcome to the podcast dialogue naturalness evaluation!**

**Task Description:**

In this test, you will listen to podcast dialogue segments generated by different systems. Your task is to evaluate the **naturalness** of the dialogue in each segment on a scale from **0 to 100**.

**Evaluation Criteria:**

- **0 – 20 (Bad):** The dialogue is completely unnatural, robotic, or awkward. It does not resemble a real conversation.
- **20 - 40 (Poor):** The dialogue has significant unnaturalness, with multiple awkward phrases, robotic tones, or inconsistent flows.
- **40 - 60 (Fair):** The dialogue is somewhat natural but has noticeable issues. It may feel rehearsed or lack smooth transitions.
- **60 - 80 (Good):** The dialogue is mostly natural, with minor unnatural elements. It resembles a real conversation but could still be improved.
- **80 - 100 (Excellent):** The dialogue sounds completely natural, like a real, spontaneous conversation between people.

**Important Notes:**

- The content of the dialogues may differ across systems. Please focus on the **overall naturalness** of the dialogue rather than the specific content or details (e.g., timbre, accent, noise or cut-off effects)
- In other words, **how realistic and similar are these dialogue segments to real podcast conversations?**
- Each test group includes a **Reference** audio extracted from a real podcast episode, representing the "**Excellent**" level of naturalness. This reference is provided to help calibrate your scoring.
- You can replay the audio segments as many times as you wish before assigning a score.
- Use headphones in a quiet environment for the best experience.

Your feedback is valuable. Thank you for participating!

Figure 7: Dialogue Naturalness Evaluation - Instruction page.

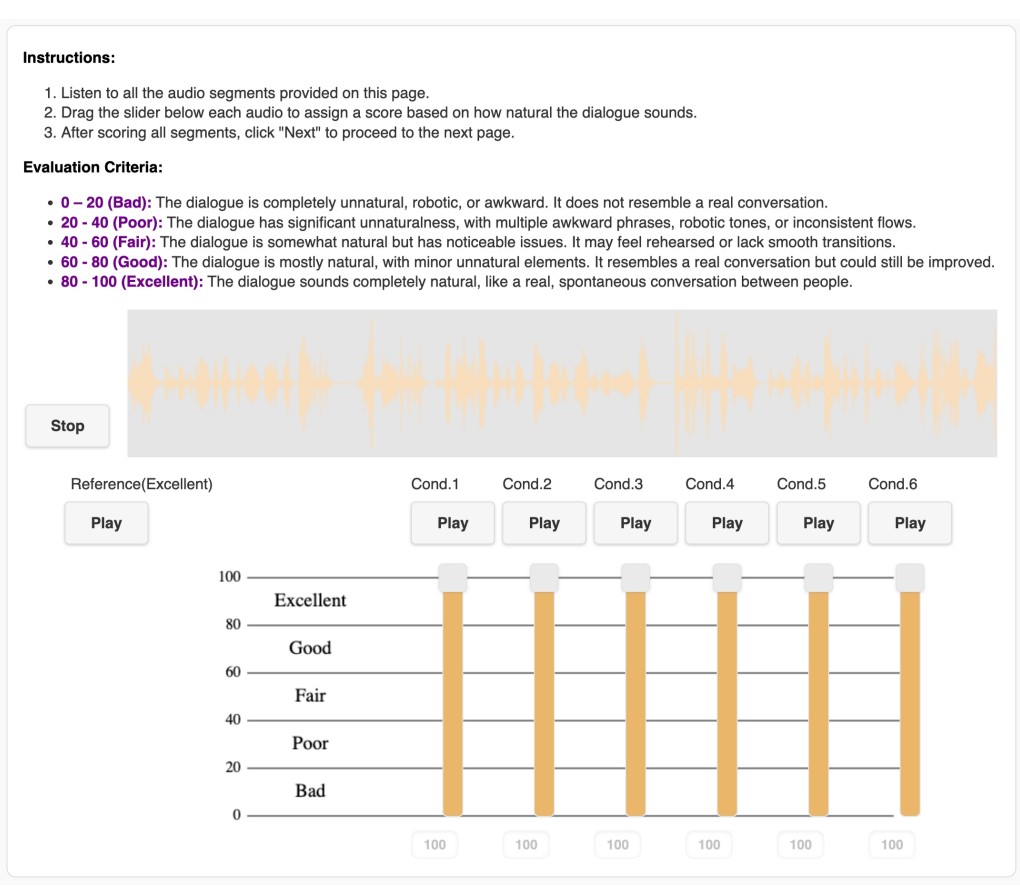

Figure 8: Dialogue Naturalness Evaluation - Test page.

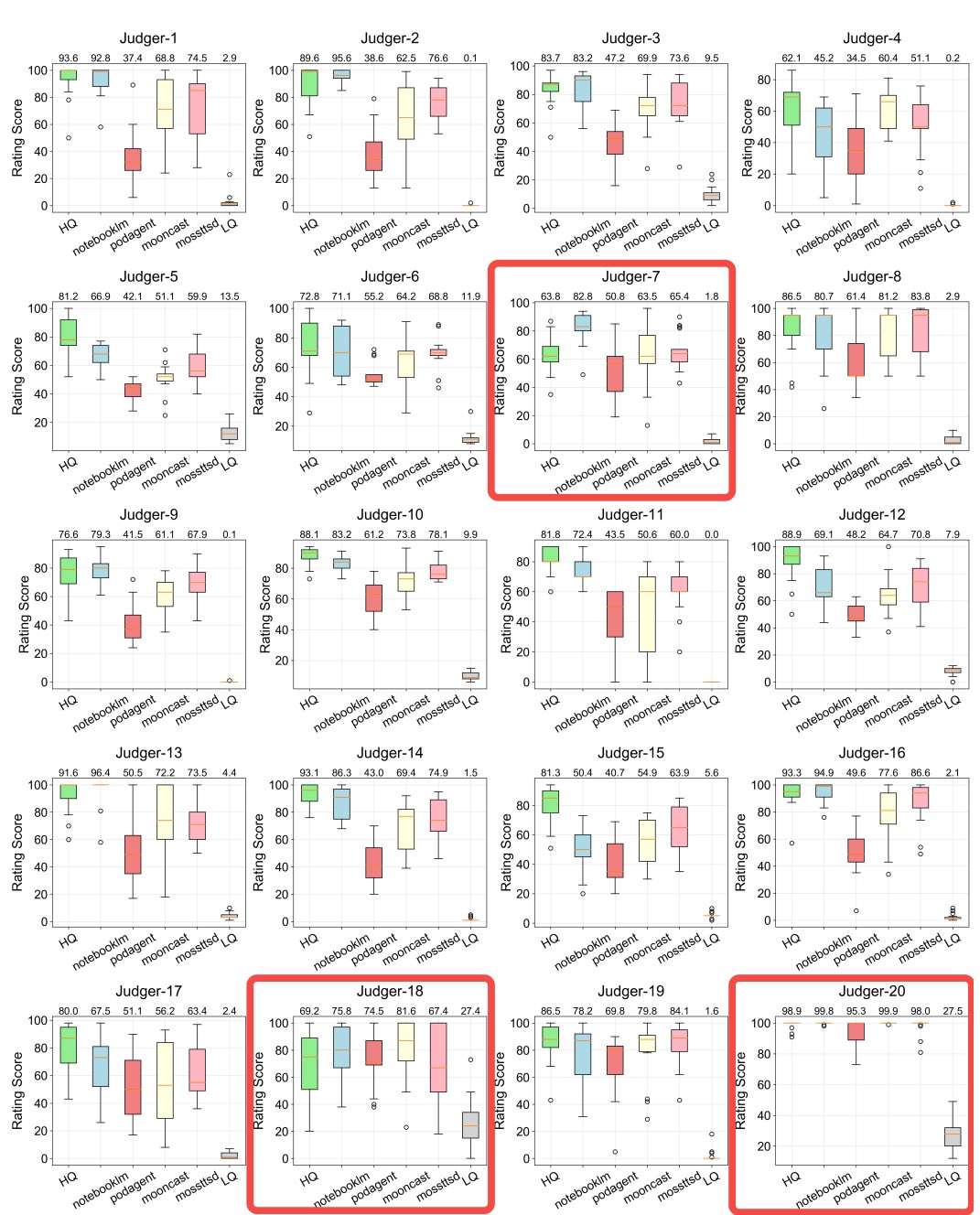

Figure 9: Dialogue Naturalness Evaluation test results from each juders.

## A.4 Audio-based evaluation (objective)

IDL: LOUD-IT; TP: LOUD-TP; LRA: LOUD-RA.

$$S_{\text{IDL}} = \begin{cases} 1, & -18 \leq \text{IDL} \leq -14, \\ e^{-k_1 \cdot (-18 - \text{IDL})}, & \text{IDL} < -18, \\ e^{-k_2 \cdot (\text{IDL} + 14)}, & -14 < \text{IDL}, \end{cases} \tag{2}$$

where $k_1$ is set as $0.0858$ to ensure $S_{\text{IDL}}$ is around $0.6$ when $\text{IDL} = -23$, and $k_2$ is set as $0.3291$ to make $S_{\text{IDL}}$ close to $0$ when $\text{IDL} = 0$.

$$S_{\text{TP}} = \begin{cases} 1, & \text{TP} \leq -1 \\ e^{-k_3 \cdot (\text{TP} + 1)}, & \text{TP} > -1 \end{cases} \tag{3}$$

where $k_3$ is set as $4.605$ to ensure $S_{\text{TP}}$ is close to $0$ when TP approaches $0$.

$$S_{\text{LRA}} = \begin{cases} 1, & 4 \leq \text{LRA} \leq 18, \\ e^{-k_4 \cdot (4 - \text{LRA})}, & \text{LRA} < 4, \\ e^{-k_5 \cdot (\text{LRA} - 18)}, & \text{LRA} > 18. \end{cases} \tag{4}$$

where $k_4$ is set as $1.1513$ to ensure $S_{\text{LRA}}$ approaches $0$ when $\text{LRA} = 0$, and $k_5$ is set as $0.2554$ to ensure $S_{\text{LRA}} \approx 0.6$ when $\text{LRA} = 20$.

Table 9: Audio-based objective metrics - Quantitative scores.

| System | LOUD_IT_SCORE | LOUD_TP_SCORE | LOUD_LRA_SCORE | SMR_BASIC_SCORE | CASP |
|---|---|---|---|---|---|
| Real-Pod | 0.72 | 0.53 | 0.82 | 0.99 | 0.58 |
| PodAgent | 0.80 | 0.32 | 1.00 | 1.00 | 0.56 |
| MoonCast | 1.00 | 0.01 | 0.68 | - | - |
| Muyan-TTS | 0.88 | 1.00 | 0.83 | - | - |
| Dia | 0.98 | 0.01 | 0.95 | - | - |
| MOSS-TTSD | 0.88 | 0.02 | 0.99 | - | - |
| NotebookLM | 0.51 | 0.56 | 1.00 | - | - |

## A.5 AUDIO-BASED EVALUATION (SUBJECTIVE)

### A.5.1 PILOT TEST

**Section 1: Quantitative Analysis (0-10 Scale)**

0 = not met at all, 5 = moderately met, 10 = fully met. Comments are optional but encouraged.

1. How well does the tone of the host or guest suit the podcast content?

| not met at all | 0 | 1 | 2 | 3 | 4 | 5 | 6 | 7 | 8 | 9 | 10 | fully met |

2. How clearly and effectively do the speakers deliver the podcast content?

| not met at all | 0 | 1 | 2 | 3 | 4 | 5 | 6 | 7 | 8 | 9 | 10 | fully met |

3. Is the speaking pace appropriate and easy to follow?

| not met at all | 0 | 1 | 2 | 3 | 4 | 5 | 6 | 7 | 8 | 9 | 10 | fully met |

4. How engaging and enjoyable is the podcast? (Does it sustain your attention throughout the episode?)

| not met at all | 0 | 1 | 2 | 3 | 4 | 5 | 6 | 7 | 8 | 9 | 10 | fully met |

5. How satisfied are you with the podcast's audio quality? (e.g., clarity, background noise)

| not met at all | 0 | 1 | 2 | 3 | 4 | 5 | 6 | 7 | 8 | 9 | 10 | fully met |

6. If background music or sound effects are present, how well do they enhance rather than interfere with the content? (Select 5 if there is no background music or sound effects)

| not met at all | 0 | 1 | 2 | 3 | 4 | 5 | 6 | 7 | 8 | 9 | 10 | fully met |

7. How likely are you to want to listen to the full episode after hearing this excerpt?

| not met at all | 0 | 1 | 2 | 3 | 4 | 5 | 6 | 7 | 8 | 9 | 10 | fully met |

**Section 2: Qualitative Analysis (YES/NO/CAN'T TELL)**

8. Does the podcast include a clear introduction and conclusion?

| YES | NO | CAN'T TELL |

9. Are background music or sound effects present in the podcast?

| YES | NO | CAN'T TELL |

10. Does the podcast sound like it was created by humans rather than AI? ("Yes" = more like humans, "No" more like AI)

| YES | NO | CAN'T TELL |

Figure 10: Questionnaire-based MOS test - Pilot test version.

### A.5.2 QUESTIONNAIRE-BASED MOS TEST

**Experiment Settings:** Lengthy listening tests can be exhausting and may lead to inaccurate feedback. It is essential to ensure the overall test duration does not exceed 30 minutes. In the Questionnaire-based MOS Test, each audio sample is around 3 minutes and requires answering 10 questions with corresponding justifications. Based on the Dialogue Naturalness Test results shown in Figure 7.2, we selected 4 representative systems. Each test group included four podcast samples from different systems but within the same podcast category. According to actual test results, each group took an average of 24 minutes to complete. The 4 representative systems are:

- **PodAgent:** An open-source podcast generation framework incorporating conversation script generation, automatic voice selection, speech synthesis, and BMSE enhancement.
- **MOSS-TTSD:** Achieved the highest score among the open-source systems utilized in the Dialogue Naturalness Evaluation (Figure 7.2).
- **NotebookLM:** A pioneering podcast generation product, widely recognized for its exceptional performance, is nearly indistinguishable from real podcasts.
- **Real-Pod:** A collection of podcasts sourced from the real world.

---

**Welcome to the Podcast Evaluation Questionnaire!**

**Study Description:**

In this study, we aim to collect **authentic feedback** on podcast audio clips. You will listen to **4 different podcast audio files**, each discussing potentially different topics. The primary goal of this research is to evaluate the **overall production quality** of the podcast segments, rather than the specific content or themes being discussed.

Each audio clip is approximately **3 minutes long** and is constructed by combining three key segments from a full podcast episode:

<The **first** minute | The **middle** minute | The **final** minute>

A brief notification sound will indicate the transitions between these segments.

**About the questionnaire:**

It consists of 8 questions, which are designed to assess the podcast audio across multiple dimensions, such as:

- Speaker's expression / Information delivery
- Audio quality / engagingness / music or sound effect harmony

**Notice:**

- We kindly ask you to **avoid** rating based on the **discussion** topic and instead focus on the requested dimension.
- Please **listen to each audio carefully**, ideally using **headphones** for optimal clarity.
- **Incomplete or insincere responses** may be subject to return. We kindly ask you to provide thoughtful and genuine feedback to ensure the effectiveness of this study.
- Please enter your **Prolific ID** as the **"Username"** in the final submission page.

Your feedback is **extremely valuable**. Thank you for your participation!

---

Figure 11: Questionnaire-based MOS test - Final version - Instruction page.

Q1. **How many speakers** are there in the podcast?

[                                        ]

Q2. How satisfied are you with the podcast's **audio quality** (e.g., clarity, volume levels, background noise)?

○ 1 = Very dissatisfied

○ 2 = Dissatisfied

○ 3 = Neutral

○ 4 = Satisfied

○ 5 = Very satisfied

Why? (Required but can be simple. The same requirement for other "Why?" questions.)

[                                                    ]

Q3. Do you like the way the guests and hosts **express** themselves?

○ 1 = Strongly dislike it

○ 2 = Dislike it

○ 3 = Neutral

○ 4 = Like it

○ 5 = Love it

Why?

[                    ]

Q4. Do you think the speakers are **effectively** delivering the information?

○ 1 = Not at all effectively

○ 2 = Not very effectively

○ 3 = Neutral

○ 4 = Somewhat effectively

○ 5 = Very effectively

Why?

[                    ]

Q5. If **music or sound effects** are present, do they **enhance or interfere** with the content? (Select Neutral if none are present)

○ 1 = Greatly interfere

○ 2 = Somewhat interfere

○ 3 = Neutral

○ 4 = Somewhat enhance

○ 5 = Greatly enhance

Why?

[                    ]

Figure 12: Questionnaire-based MOS test - Final version (Question 1-5).

Q6. How **engaging** is the podcast?

◯ 1 = Not engaging at all

◯ 2 = Slightly engaging

◯ 3 = Neutral

◯ 4 = Engaging

◯ 5 = Extremely engaging

Why?

[                    ]

Q7. How likely are you to listen to the **full episode** after hearing this?

◯ 1 = Not likely at all

◯ 2 = Slightly likely

◯ 3 = Neutral

◯ 4 = Likely

◯ 5 = Very likely

Why?

[                    ]

Q8. Does the podcast sound like it was created by **humans** rather than **AI**?

◯ 1 = Definitely AI

◯ 2 = More like AI

◯ 3 = Neutral -- Could be either human or AI

◯ 4 = More like humans

◯ 5 = Definitely humans

Why?

[                    ]

Q9. (Optional) Any additional comments on this podcast audio?

[                              ]

Figure 13: Questionnaire-based MOS test - Final version (Question 6-9).

Table 10: Questionnaire-based MOS test - (Q.) represents the average score from the direct scoring answers, and (J.) represents the score derived from the justifications.

| Systems / Metrics | MOSS-TTSD | | NotebookLM | | PodAgent | | Real-Pod | |
|---|---|---|---|---|---|---|---|---|
| | Q. | J. | Q. | J. | Q. | J. | Q. | J. |
| Information Delivery | 4.0 | 3.0 | 4.2 | 4.0 | 1.6 | 1.0 | 4.2 | 4.0 |
| Music/Sound Effects | N/A | N/A | N/A | N/A | 2.4 | 2.0 | 3.3 | 3.0 |
| Engagement Level | 2.2 | 3.0 | 3.1 | 3.0 | 1.1 | 1.0 | 3.6 | 4.0 |
| Full Episode Likelihood | 2.1 | 2.0 | 2.1 | 3.0 | 1.0 | 1.0 | 2.3 | 3.0 |
| Human Likelihood | 3.0 | 3.0 | 3.3 | 3.5 | 1.1 | 1.0 | 4.2 | 4.0 |
| Audio Quality | 3.5 | 3.0 | 4.2 | 4.0 | 3.0 | 2.0 | 3.9 | 4.0 |
| Speaker Expression | 3.3 | 3.0 | 4.0 | 3.0 | 1.5 | 1.0 | 3.4 | 4.0 |

## A.6 SYSTEM ANALYSIS REPORT

### PodAgent

PodAgent is an open-source podcast generation framework that integrates conversation script generation, automatic voice selection, speech synthesis, and music/sound effects (MSE) enhancement.

**Strengths**

- **Comprehensive Automation**
  PodAgent supports a multi-agent system ("Host-Guest-Writer") that generates informative conversation scripts and automates voice selection, making it a versatile tool for podcast creation.
- **Music/Sound Effects Integration**
  PodAgent performs well in MSE-related metrics, such as **Speech-to-Music Ratio (SMR)** and **CASP (MSE-Speech Harmony)**. While it does not match the upper limits of human-made podcasts, its results are consistent, making it a practical alternative for efficient audio program creation.
- **Objective Speech Quality**
  PodAgent demonstrates competitive performance in objective metrics like DNSMOS (speech quality), showcasing its ability to produce clear and intelligible speech.
- **Open-Source Advantage**
  Being open-source, PodAgent is accessible for public use and research, enabling further refinement and experimentation.

**Weaknesses**

- **Dialogue Naturalness**
  PodAgent scored poorly in subjective dialogue naturalness tests, attributed to its reliance on a single-sentence synthesis TTS system (CosyVoice2).
- **Speaker Similarity**
  In terms of **Speaker Similarity (SIM)**, PodAgent underperformed compared to other systems. Its instruction-following style control strategy sacrifices vocal fidelity to enhance conversational expressiveness.
- **Short Scripts**
  The evaluation shows that PodAgent-generated podcast scripts lack the richness and diversity of real-world podcasts, which are typically longer and more information-dense.

**Potential Improvements**

- **Upgrade to Dialogue Synthesis TTS**: Transitioning to a multi-sentence or dialogue-level TTS system could significantly improve naturalness and interactivity.
- **Enhance Speaker Similarity**: Refining the instruction-following mechanism to improve speaker similarity for better timbre consistency in the long-form podcast.
- **Longer Script Generation**: Developing methods to handle longer, richer scripts could close the gap with real podcasts in terms of content diversity and informativeness.

Figure 14: System analysis report based on PodEval - PodAgent.

## A.7 WSIM AMONG DIFFERENT SYSTEMS

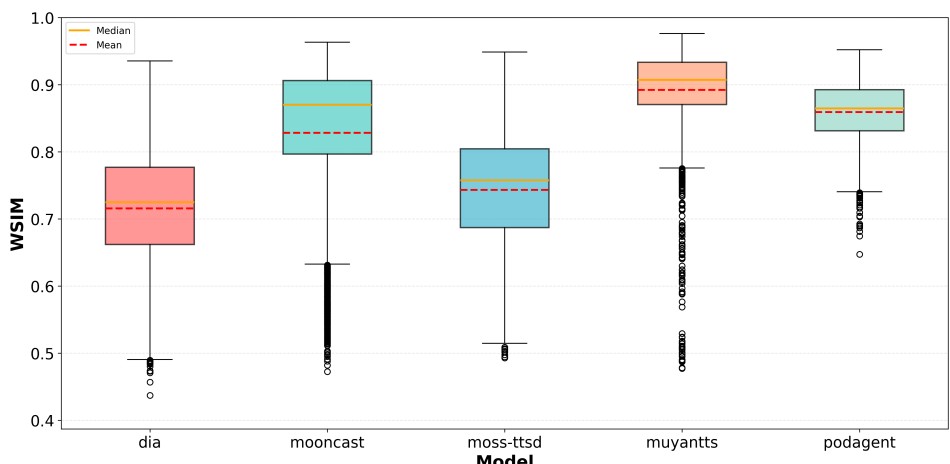

Figure 15: WSIM among different systems.

## A.8 INTER RATER RELIABILITY

Figure 16: Pearson Correlation among Different Human Judgers.

