# OpenReview forum: "PodEval: A Multimodal Evaluation Framework for Podcast Audio Generation"
_ICLR.cc/2026/Conference — Submitted to ICLR 2026_

### Official Review · Reviewer_13hU · 2025-10-30

**Soundness:** 3
**Presentation:** 4
**Contribution:** 3
**Rating:** 8
**Confidence:** 4

**Summary:**

This paper proposes PodEval, a comprehensive, open-source evaluation framework designed for multimodal, long-form podcast audio generation. The work addresses the significant challenge of evaluating open-ended generative tasks, where there is no single ground-truth reference. The framework decomposes the evaluation into three core dimensions—text (conversation content), speech (spoken dialogue), and audio (overall sound including music, sound effects, and their integration)—with a clear distinction between "Content" and "Format" quality. To support this, the authors construct a real-world podcast dataset ("Real-Pod") spanning diverse topics, serving as a reference for human-level creative quality. For each dimension, PodEval provides a suite of evaluation methods: a combination of quantitative metrics and LLM-as-a-judge for text; objective metrics (WER, SIM, DNSMOS) and a subjective Dialogue Naturalness test for speech; and objective metrics (Loudness, SMR, CASP) along with a novel Questionnaire-based MOS test for audio. The framework is validated through experiments on a range of podcast generation systems, and the entire project is released as open-source.

**Strengths:**

Highly Practical and Timely Contribution: The paper addresses a critical and under-served problem in the AIGC community: the evaluation of open-ended, long-form generative content. With the rise of AI-generated podcasts and similar content, a standardized evaluation framework is desperately needed. PodEval provides a well-structured, practical solution that researchers can immediately adopt, making it a valuable community resource.

Comprehensive and Well-Designed Evaluation Methodology: The decomposition of the task into text, speech, and audio dimensions is clear and logical. The framework's strength lies in its combination of multiple evaluation strategies. The use of LLM-as-a-judge for text content, the adaptation of the MUSHRA framework for a Dialogue Naturalness test with attention checks, and the novel approach of concatenating beginning/middle/end segments for long-form audio MOS testing are all well-considered and effective design choices that address the core challenges of reliability and scalability.

Creation of a Valuable Benchmark Dataset: The construction of the "Real-Pod" dataset is a significant contribution in itself. By curating a diverse collection of real-world podcasts across 17 categories, the authors provide a high-quality reference point for human-level quality. This dataset serves as a crucial benchmark for future work in podcast generation and evaluation, enhancing the long-term value of this paper.

**Weaknesses:**

While the framework is novel and well-integrated, many of the individual evaluation techniques are adaptations of existing methods. For instance, using LLMs as judges (e.g., MT-Bench), DNSMOS for speech quality, and FAD for audio quality are well-established practices. The paper's primary contribution is the curation and combination of these methods rather than the invention of fundamentally new evaluation algorithms.

**Questions:**

The paper describes a manual review process for curating the Real-Pod dataset but fails to specify the number of human experts involved or their qualifications, leaving the potential for bias and subjectivity in the benchmark's creation unaddressed.

---

> ### Author Response · Authors · 2025-11-20
> **Response to Reviewer 13hU**
>
> Dear Reviewer,
>
> We sincerely thank you for the **thoughtful and encouraging feedback**! We are truly delighted that you recognized the *Highly Practical and Timely Contribution* of this work. Many of the aspects you highlighted — such as the *multi‑dimensional evaluation design*, *the integration of both objective and subjective assessments*, and *the construction of the Real‑Pod dataset* — deeply resonate with our original goals. It means a lot to us that you see the value and motivation behind our efforts to build a **comprehensive and practical framework** for evaluating *open‑ended, long‑form generative content*.
>
> Regarding your question about the **curation of the Real‑Pod dataset**, we completely agree that *transparency* is essential for a fair benchmark. In our curation process, *three human experts* — with backgrounds in speech and audio processing, generation, and extensive experience as podcast enthusiasts — manually filtered and verified the data to ensure quality and diversity. We will make these details clearer in the final version to enhance both transparency and reproducibility.
>
> We sincerely appreciate your recognition and constructive comments, which greatly encourage us to further refine this work and make PodEval an even more reliable and valuable resource for the community. Thank you again, and we would be very happy to discuss any further questions or suggestions you might have.

---

### Official Review · Reviewer_d89n · 2025-10-30

**Soundness:** 2
**Presentation:** 3
**Contribution:** 2
**Rating:** 2
**Confidence:** 4

**Summary:**

This paper introduces PodEval, a comprehensive evaluation framework for the open-ended task of podcast audio generation. The authors identify key challenges in this domain, such as the lack of reference standards and the difficulty of evaluating long-form, multi-modal content. To address this, they propose a structured approach that decomposes the evaluation into three dimensions: text, speech, and audio. For each dimension, a suite of objective metrics and subjective listening tests are designed. A key contribution is the creation of the Real-Pod dataset, a curated collection of real-world podcasts to serve as a human-level quality reference. The framework is demonstrated by evaluating several podcast generation systems, including open-source, closed-source, and human-created examples.

**Strengths:**

1. Problem Significance: The paper tackles the highly relevant and difficult problem of evaluating open-ended, long-form audio generation, a critical bottleneck for progress in this area.
2. Comprehensive Structure: The decomposition of the evaluation into text, speech, and audio dimensions, with a further distinction between "Content" and "Format," provides a logical and structured way to approach this complex task.
3. Valuable Dataset: The Real-Pod dataset is a well-curated and useful resource. The process for its creation is transparent and methodologically sound, providing a much-needed benchmark for human-level quality.
4. Rigorous Subjective Test Design: The subjective evaluation methodologies, particularly the Dialogue Naturalness test based on MUSHRA with high/low-quality anchors and the detailed questionnaire-based MOS test, are well-designed and incorporate best practices like spammer detection to ensure data validity.

**Weaknesses:**

1. Contradiction Between Objective and Subjective Results: This is the most significant weakness of the paper. The PodAgent system, which appears to be a focal point of the evaluation, consistently scores well on many of the proposed objective metrics (e.g., high SPTD in Fig 3, perfect SMR_SCORE and good CASP in Table 9 and Fig 5), yet it performs very poorly in the subjective tests for dialogue naturalness (Fig 4) and overall listening experience (Fig 6). An evaluation framework is only effective if its objective metrics are meaningful proxies for human perception. This stark discrepancy suggests that the proposed objective metrics are either insufficient or are measuring the wrong signals, thus failing to capture what actually constitutes a high-quality podcast.
2. Potential for Evaluation Bias: The paper heavily features PodAgent. The objective metrics seem to align particularly well with the architectural strengths of PodAgent (e.g., distinct speaker timbres leading to high SPTD, clear separation of speech and music leading to good SMR/CASP). However, the system's core weakness—its reliance on a single-sentence TTS model, leading to unnatural dialogue flow—is only revealed by the subjective tests. This raises concerns that the framework may be unintentionally tailored to a specific type of system, diminishing its generalizability and fairness as a benchmark. The framework's value is diminished if it rewards systems for being objectively "correct" on certain technical measures while failing on the ultimate goal of perceptual quality.
3. Lack of Correlation Analysis: The paper presents objective and subjective results separately but provides no quantitative analysis of the correlation between them. For an evaluation paper, demonstrating a strong correlation between proposed automatic metrics and human judgments is paramount to validating the metrics. Without this analysis, the utility of the objective part of PodEval remains unproven.

**Questions:**

1. The core issue is the discrepancy between objective and subjective scores for systems like PodAgent. How should a user of PodEval interpret a scenario where a system excels on objective metrics but fails catastrophically in human evaluation? Does this not invalidate the utility of the objective metrics for predicting real-world performance?
2. Could the authors provide a correlation analysis (e.g., Pearson or Spearman correlation) between the scores from the objective metrics (especially the novel ones like SPTD, and audio-related ones like SMR and CASP) and the subjective scores (Dialogue Naturalness, Human Likelihood, etc.) across all tested systems? This would be the most direct way to validate the claim that the framework is effective.
3. Given that PodAgent's poor subjective performance is attributed to its single-sentence TTS backend, it seems dialogue-level prosody and coherence are the most critical factors for naturalness. None of the proposed objective metrics appear to directly measure this. Do the authors believe a new objective metric is needed to capture this crucial aspect, and if so, why was it not included in the PodEval framework?
4. What steps were taken during the design of the objective metrics to ensure they were general and not biased towards the specific outputs of the PodAgent system?

---

> ### Author Response · Authors · 2025-11-20
> **Response to Reviewer d89n**
>
> Dear Reviewer,
>
> Thank you for your time and effort in reviewing our manuscript. We are grateful for your positive assessment of our contributions, including that *“the paper tackles a highly relevant and difficult problem of evaluating open‑ended, long‑form audio generation”*, *“the decomposition into text, speech, and audio dimensions with ‘Content’ and ‘Format’ sub‑dimensions provides a logical evaluation structure”*, *“the Real‑Pod dataset is well‑curated and methodologically sound”*, and *“the subjective evaluation design is rigorous and reliable.”*
>
> We equally appreciate your constructive questions, which represent invaluable opportunities to enhance the clarity and impact of our work. Here we would like to address these points and try to avoid any misunderstanding:
>
> ---
>
> ### Q1: Correlation between Objective and Subjective Results
>
> **A1:** We would like to clarify that in our framework, **objective and subjective evaluations are designed to be complementary and mutually reinforcing, rather than substitutive**. Evaluating *open‑ended, long‑form audio generation* is a highly challenging task, and it is unrealistic to expect a single objective metric to fully capture human perception across all dimensions of quality.
>
> The **objective metrics** in PodEval mainly target **well‑defined and measurable indicators grounded in widely accepted prior knowledge** from the speech and audio research community. For example, WER can *quantitatively assess model’s robustness*, SIM and SPTD are built upon extensive work in *speaker representation learning*. In contrast, the **subjective metrics** focus on **higher‑level perceptual** dimensions such as *dialogue naturalness* and *overall listening experience*, which inherently involve human interpretation and context understanding.
>
> Therefore, the observation that a system like *PodAgent* performs well on certain objective metrics yet poorly on some subjective ones **does not indicate a contradiction**. Instead, it validates the need for a **multidimensional evaluation**. Our framework effectively captures **complementary facets of long‑form audio quality**, rather than relying on a single, coarse measurement.
>
> ---
>
> ### Q2: Potential for Evaluation Bias？
>
> **A2:** We respectfully **disagree** with the concern that the framework may be biased toward the *PodAgent* system. **In fact, the design of our evaluation explicitly avoids such bias.** If the framework were intended to favor PodAgent, there would be no reason for us to invest extensive effort into developing and conducting rigorous *subjective listening tests*, where PodAgent actually performs poorly. On the contrary, the inclusion of these human evaluations is precisely what ensures fairness and comprehensiveness across different system types.
>
> Moreover, as we have emphasized in the paper, **objective and subjective evaluations serve complementary purposes**. The objective metrics capture well defined and measurable properties, whereas the subjective tests reveal perceptual feedback—particularly crucial for *open‑ended long‑form audio evaluation*. This review comment seems to imply that objective metrics should dominate the assessment of model quality, which differs from our intended design. Our framework aims to integrate both **measurement‑driven** and **perception‑driven** dimensions, ensuring that no single system type is advantaged or penalized due to its architectural characteristics.

---

> > ### Author Response · Authors · 2025-11-20
> > **Response to Reviewer d89n**
> >
> > ### Q3: Objective metrics for dialogue-level prosody and coherence
> >
> > **A3:** We agree that designing objective metrics that can directly reflect human perception of speech—such as naturalness—is undoubtedly valuable. There are studies specifically devoted to training models to **approximate human judgment scores**, aiming to build learnable perceptual metrics. However, at present, most reliable and interpretable objective measures still rely on having a **standard reference** for comparison. For **naturalness**, especially at the **dialogue level**, no existing objective method can yet fully replace human evaluation.
> >
> > We envision integrating such advanced perceptual metrics into our framework in the future. However, in this work, our primary goal is to establish a **comprehensive and practically applicable evaluation framework** that aligns closely with **real‑world user needs**, while ensuring that its evaluation results are reliable and usable in practice. To achieve this, we have devoted significant effort to the **design of subjective evaluation protocols**.
> >
> > As we mentioned above, in our framework, **objective and subjective evaluations** are **complementary and mutually reinforcing, rather than substitutive**. For example, metrics such as *WER* can quantitatively assess model robustness. Only when a system reaches sufficient stability does subjective evaluation of **naturalness** become truly meaningful. This progressive relationship not only enhances evaluation reliability but also **reduces the human evaluation cost**.
> >
> > Importantly, evaluating **open‑ended long‑form audio** is inherently a **challenging task**, even from a human listener’s perspective. This further underscores the necessity of establishing an **effective, user‑centric evaluation framework** that can enable the collection of **richer and more reliable human‑preference data**, ultimately guide the development of future **automated naturalness evaluation methods**.
> >
> > ---
> >
> > We hope our response has thoroughly addressed your questions and concerns. We would be happy to discuss any additional points you may have and would greatly appreciate the opportunity for you to reconsider your evaluation based on this clarification.

---

> > > ### Comment · Reviewer_d89n · 2025-11-27
> > >
> > > I thank the authors for their detailed responses and the additional analyses (e.g., WSIM and correlation data).
> > >
> > > I have raised my score because the authors successfully addressed my concerns regarding potential evaluation bias. The clarification that the framework exposes the disconnect between PodAgent's acoustic consistency and its prosodic unnaturalness is valid. I also acknowledge the practical contribution of the Real-Pod dataset and the open-source pipeline, which will be useful for the community.
> > >
> > > However, I align with Reviewer 6Cwy regarding the limited methodological novelty. The framework largely aggregates existing metrics (WER, SIM, DNSMOS) rather than proposing novel evaluation methodologies tailored for the specific challenges of long-form audio (e.g., narrative coherence, long-term prosody). While I accept the argument that objective and subjective metrics are "complementary," the ultimate goal of an automated evaluation framework is to serve as a reliable proxy for human perception. The fact that a system can score perfectly on objective metrics (like SPTD/SMR) but fail subjectively limits the framework's utility for rapid, automated model iteration. The correlation of r=0.53 for SPTD, while positive, suggests it is a somewhat weak signal for quality.
> > >
> > > The paper presents a solid engineering effort and a valuable resource, but the scientific novelty and the predictive validity of the proposed objective metrics are marginally below the bar for a top-tier conference like ICLR.

---

> ### Author Response · Authors · 2025-11-28
> **Response to Reviewer d89n**
>
> **Dear Reviewer:**
>
> Thank you very much for your follow‑up comments and for raising your score after reviewing our additional analyses. We particularly appreciate your acknowledgment of **(1)** our successful mitigation of evaluation‑bias concerns, and **(2)** the significant practical value of the Real‑Pod dataset and open‑source pipeline for the broader research community.
>
> We respectfully address the remaining concerns:
>
> ### Q1. Methodological novelty
> **A1.** Methodological or scientific novelty can be somewhat subjective. If the concern arises simply because our framework incorporates several existing objective metrics, we respectfully disagree. These metrics constitute only one component of the framework, and our work includes several additional innovations and contributions. Although we have already addressed these points elsewhere, we summarize the key contributions again here:
>
> - **Our key innovation is a principled decomposition of long‑form audio evaluation into interpretable components.** Because long‑form audio spans diverse factors, from linguistic content to acoustic naturalness and dialogue dynamics, it cannot be assessed with a single metric. We address this by separating the problem into two fundamental perspectives, content and form, operationalized through a unified text‑speech‑audio three‑dimensional evaluation framework.
>
> - Because open‑ended generation lacks a single ground‑truth reference, we **build a real‑world podcast dataset covering diverse categories and topics**. This collection serves as a human‑level creative benchmark, providing essential reference points for evaluation and enabling meaningful comparisons between AI‑generated and human‑produced content.
>
> - We design a **comprehensive subjective evaluation framework to ensure reliable and interpretable human judgments**. It includes dialogue‑level naturalness assessments, multi‑dimensional ratings of long‑form audio (e.g., engagement and information delivery), and newly structured evaluation procedures, including a MUSHRA‑based setup for dialogue naturalness and a questionnaire‑based approach for overall audio quality. The framework also standardizes data formatting for variable‑length outputs and introduces dedicated attention checks to maintain rating robustness.
>
> - The **objective metrics** in PodEval focus on *well‑defined, measurable indicators grounded in widely accepted knowledge within the speech and audio research community*. Some of these metrics are **newly** proposed, while others are established. For the established ones, our contribution lies in bridging the gap between “having a metric” and **“making it practically applicable”** to long‑form podcast evaluation. We adopt these metrics because they provide effective assessment along certain dimensions, which is beneficial for the overall framework. *We cannot discard proven, reliable indicators solely for the sake of novelty, especially when they comprise only a small portion of the entire system.* **Their inclusion does not diminish the overall methodological novelty of our work.**
>
> ---
>
> ### Q2. Predictive validity of SPTD
> **A2.**  A Pearson correlation coefficient of **r = 0.53 (p < 0.0001)** is considered a *LARGE EFFECT SIZE* [1], reflecting a *moderate‑to‑strong* positive correlation that is statistically significant.
>
> Furthermore, when evaluating whether a metric is “good,” it is important not to interpret the absolute value in isolation but to consider the **task setting** and **established correlation levels in related research**. For example, in [2], which studies the relationship between acoustic similarity and human‑perceived voice similarity, a task highly relevant to SPTD, the reported correlation is 0.33, indicating a moderate association between perceptual similarity and acoustic similarity. Based on this reference value, *the SPTD correlation of r = 0.53 (p < 0.0001) can be considered a reasonably strong result for this task*.
>
>
> [1] Cohen J. Statistical power analysis for the behavioral sciences[M]. routledge, 2013.
>
> [2] Liu S, Babel M, Zhu J. A comparison of voice similarity through acoustics, human perception and deep neural network (DNN) speaker verification systems[C]//Proc. Interspeech. 2024: 3674-3678.
>
> ---
> Thank you once again for taking the time to provide thoughtful follow‑up comments and raise your score. We hope that our additional responses have fully addressed your questions and concerns. Should there be any remaining points that would benefit from further clarification, we would be delighted to provide additional details. We deeply appreciate your careful consideration of our work and would be honored if you found our responses merit further positive consideration.

---

### Official Review · Reviewer_WWEy · 2025-11-01

**Soundness:** 3
**Presentation:** 3
**Contribution:** 3
**Rating:** 6
**Confidence:** 4

**Summary:**

This paper introduces PodEval, a multimodal evaluation framework for podcast-like long-form audio generation. The authors argue that existing multimodal benchmarks mostly focus on understanding tasks, whereas open-ended generative evaluation, especially for long-form audio, is underexplored. PodEval decomposes the evaluation into three dimensions: text, speech, and audio, each assessed through both objective metrics (e.g., BLEU, DNSMOS, CASP) and subjective listening tests. They also curate a real-world dataset called Real-Pod, spanning 17 podcast categories and 51 topics, to serve as a human-quality reference. The framework includes both quantitative
metrics (Distinct-N, Info-Density, SPTD, etc.) and LLM-as-a-judge evaluations. Experimental results compare various open-source, closed-source, and human-created podcast systems (PodAgent, MoonCast, NotebookLM, etc.) and demonstrate the reliability of PodEval as a standardized evaluation pipeline.

**Strengths:**

• Tackles an underexplored but important problem: evaluating open-ended long-form multimodal generation.

• Provides a clear multimodal decomposition (text/speech/audio) with both objective and subjective evaluation methods.

• Introduces the SPTD metric for speaker timbre diversity and validates it empirically.

• Comprehensive dataset (Real-Pod) and open-source toolkit increase reproducibility and potential community impact.

• Thoughtful subjective test design with attention checks and justification mechanisms.

**Weaknesses:**

• The work’s novelty is primarily integrative rather than conceptual; it reuses many established metrics and evaluation settings.

• The proposed SPTD metric lacks deeper theoretical analysis or correlation studies with human perception.

• Subjective tests use a relatively small participant pool (n=20), which may limit generalizability.

• Some sections (especially methodology details) are overly descriptive and could be summarized more succinctly.

• The domain focus on podcasts might limit transferability to other long-form generative modalities (e.g., audiobooks or radio).

**Questions:**

• How robust is the SPTD metric across different TTS models and languages? Does it correlate well with human-perceived speaker distinction?

• Could the authors generalize PodEval to other long-form generative audio types (e.g., storytelling or news broadcast)?

• Are there any findings on inter-rater reliability in the subjective listening tests?

• How might LLM-as-a-judge results change if newer audio-language models are used instead of GPT-4?

---

> ### Author Response · Authors · 2025-11-20
> **Response to Reviewer WWEy**
>
> Dear Reviewer,
>
> Thank you for your valuable time invested in evaluation of our manuscript and providing detailed feedback. We are encouraged by your recognition of our contributions, including *"tackling an underexplored yet important problem"*; *"proposing a clear multimodal decomposition (text, speech, and audio) with both objective and subjective evaluation methods"*; *"introducing metrics and validating them empirically"*; *"constructing a comprehensive real‑world dataset and open‑source toolkit that enhance reproducibility"*; and *"designing a thoughtful and reliable subjective evaluation"*.
>
> We equally appreciate your constructive questions, which represent invaluable opportunities to enhance the clarity and impact of our work. Here we would like to address these points and try to avoid any misunderstanding:
>
> ---
>
> ### Q1: Discussion on reusing existing metrics
>
> **A1:** Open-ended long-form audio evaluation is a highly challenging task, although we leveraged some existing metrics in our evaluation framework, the contribution of this work goes far beyond that, for example:
>
> 1. **Conceptual decomposition of a complex task.** A main innovation of our approach is to decompose the task into multiple interpretable dimensions. It is unrealistic to evaluate long‑form audio with a single metric. Such audio involves many different aspects — linguistic informativeness, acoustic naturalness, speech stability, dialogue dynamics, listener perception and so on. Therefore, we decompose this complex task into two fundamental perspectives, **“content”** and **“form”**, which are implemented through our systematic **text‑speech‑audio** three‑dimensional evaluation framework.
>
> 2. Since it is an **open‑ended generation task**, there is no single ground‑truth reference available. We construct a **real-world podcast reference dataset** spanning a wide range of podcast categories and topics, serving as a reference for human-level creative quality. This dataset provides essential ground truth for evaluation and enables meaningful comparison between AI-generated and human-created content.
>
> 3. We carefully design a comprehensive **subjective evaluation framework** to ensure **reliable and interpretable** human assessments. The framework includes:
>
>    a. **Naturalness assessment** for dialogue‑level speech segments.
>    b. **Multi‑dimensional feedback** on overall long‑form audio, covering engagement, information‑delivery effectiveness, willingness to listen to the full episode, and other key aspects.
>    c. **Strategic selection of evaluation methods**:
>       - For *dialogue naturalness*, we adopt a *MUSHRA‑based testing framework* and design reference, high‑quality, and low‑quality samples for pairwise comparison, which improves the reliability and confidence of subjective judgments.
>       – For *overall audio evaluation*, we use a *questionnaire‑based assessment* that balances coverage of essential dimensions and practical time constraints.
>    d. *Unified data formatting* to accommodate variable‑length, open‑ended audio generations.
>    e. *Attention‑check mechanisms* to verify data validity and ensure the robustness of human ratings.
>
> 4. For those established metrics, we bridge the gap between "having a metric" and **"making it practically applicable"** for long-form podcast evaluation. For example, Speaker Similarity (SIM) is commonly used in zero-shot TTS evaluation, but standard applications simply compute similarity between reference and target audio. In the podcast evaluation context, to make the evaluation framework applicable for any podcast generation system, we deal with the final complete podcast audio rather than the intermediate speech segments. Therefore, we design a process pipeline: background sound removal → speaker boundary detection → segment extraction → same-speaker clustering → SIM computation. During the process, we further optimize the effectiveness by filtering short segments that providing insufficient suprasegmental information for accurate speaker representation.
>
> In summary, our work goes beyond reusing existing metrics — it establishes a comprehensive **conceptual** and **practical** framework for **open‑ended long‑form audio evaluation**. Moreover, all components — including data processing workflows, objective metric implementations, and the web interface for subjective evaluations — have been **fully open‑sourced**. While some implementation details may not be fully presented in the paper due to space limitations, we hope the above discussion helps to **clarify our contributions** and address your concerns.

---

> > ### Author Response · Authors · 2025-11-20
> > **Response to Reviewer WWEy**
> >
> > ### Q2: Correlation of SPTD with human perception
> >
> > **A2:** SPTD is implemented by calculating the average dissimilarity among speaker embeddings, and therefore its performance depends on the effectiveness of **speaker representation** used. In our Experiment, we employed the open‑source pyannote [1][2] to extract embeddings. Specifically, this model achieves 12.2% DER (diarization error rate) on the AISHELL‑4 dataset (Chinese) and 11.3% DER on VoxConverse(v0.3, English).
> >
> > We also tried to investigate its **correlation with human-perceived speaker distinction**. Specifically, we evaluated the correlation between the objective SPTD metric and subjective human ratings from 10 evaluators. The overall Pearson correlation coefficient was **r = 0.53 (p < 0.0001)**, indicating a strong positive correlation.
> >
> > [1] Bredin, Hervé. *"pyannote. audio 2.1 speaker diarization pipeline: principle, benchmark, and recipe."* 24th INTERSPEECH Conference (INTERSPEECH 2023). ISCA, 2023.
> > [2] Plaquet, Alexis, and Hervé Bredin. *"Powerset multi-class cross entropy loss for neural speaker diarization."* arXiv preprint arXiv:2310.13025 (2023).
> >
> > ---
> >
> > ### Q3: Transferability to other long-form audio types
> >
> > **A3:** Yes, absolutely. While PodEval was developed around the podcast scenario, its **text–speech–audio multidimensional evaluation framework** is easily transferable to other long‑form audio types.
> >
> > For example, in the case of **audiobooks**, the text content is fixed and usually does not require textual‑level evaluation. The main challenge lies in the **long‑form audio evaluation**. Therefore, PodEval’s *speech‑based* and *audio‑based* evaluation modules can be directly applied.
> >
> > - **Speech‑based evaluation:** All objective metrics remain valid. The subjective tests can simply replace dialogue clips with audiobook segments while using professionally recorded excerpts as reference and high‑quality anchors. The focus stays on *naturalness*.
> > - **Audio‑based evaluation:** All objective metrics are also applicable. The three‑segment long‑audio subjective design and the questionnaire framework can be largely reused, except that *information‑delivery effectiveness* is less relevant in the audiobook scenario and can be omitted.
> >
> > Moreover, all **attention check** and **justification mechanisms** designed for the subjective tests can be reused to ensure data reliability across different long‑form audio domains.
> >
> > ---
> >
> > ### Q4: Inter-rater reliability in the subjective listening tests
> >
> > **A4:** Thank you for the suggestion. We computed the Intraclass Correlation Coefficient (**ICC(2)**, two‑way random‑effects model) [3] to assess the inter‑rater reliability before and after applying our attention‑check strategy. The results are summarized below:
> >
> > | Condition | ICC (2) |
> > |------------|----------|
> > | Before | 0.6861 |
> > | After  | 0.7391 |
> >
> > The increase in ICC demonstrates that the attention‑check mechanism effectively **enhanced rating consistency** across listeners. The final **ICC** is **0.7391 (p < 0.001)**, which is considered **good reliability**. The **average Pearson correlation** across raters was **0.79**, confirming that raters ranked stimuli in a **consistent** manner. (The Pearson correlation heatmap is attached in **Appendix A.8 in the revised PDF**.)
> >
> > [3] P. E. Shrout and J. L. Fleiss, *Intraclass correlations: Uses in assessing rater reliability*, *Psychological Bulletin*, 86 (2), 420–428, 1979.

---

> > > ### Author Response · Authors · 2025-11-20
> > > **Response to Reviewer WWEy**
> > >
> > > ### Q5: How might LLM-as-a-judge results change if newer audio-language models are used instead of GPT-4?
> > >
> > > **A5:** That’s an insightful question! We were also curious whether the latest audio-enabled LLMs could perform naturalness judgments effectively. To explore this, we evaluated two state-of-the-art audio LLMs from Google, **Gemini 2.5‑Flash** and **Gemini 2.5‑Pro**, in the *dialogue naturalness evaluation* task on podcast segments. Here are example results:
> > >
> > > | Model | LQ | HQ | MoonCast | MOSS-TTSD | NotebookLM | PodAgent |
> > > |--------|----|----|-----------|------------|-------------|-----------|
> > > | Gemini‑2.5‑Flash | 85 | 80 | 75 | 90 | 88 | 82 |
> > > | Gemini‑2.5‑Pro   | 15 | 70 | 90 | 95 | 88 | 92 |
> > >
> > > We observe that **Gemini‑2.5‑Flash** assigns an unexpectedly high score (85) to the *LQ* system output, which was generated using *eSpeak*—a synthetic voice that sounds extremely robotic and is trivial for human judges to detect.
> > >
> > > In contrast, **Gemini 2.5‑Pro** shows a better overall alignment with human perception, assigning much lower scores to clearly robotic speech. However, it still rated high‑quality (HQ) human podcast segments only around 70, while other advanced synthetic systems received scores around 90.
> > >
> > > These results suggest that **even the most advanced audio LLMs struggle with fine‑grained naturalness evaluation**, especially in conversational contexts where human‑level perceptual sensitivity is essential.
> > >
> > > ---
> > >
> > > Thank you for your insightful feedback and recognition. We hope that our responses have addressed your concerns and clearly demonstrated the potential impact of our work. Your suggestions have been very helpful in improving both the clarity and rigor of our study.  We sincerely welcome any further questions or suggestions you may have.

---

### Official Review · Reviewer_6Cwy · 2025-11-01

**Soundness:** 3
**Presentation:** 3
**Contribution:** 2
**Rating:** 4
**Confidence:** 3

**Summary:**

PodEVAL proposes a multimodal evaluation framework specifically for systems that generate long-form dialogue in a podcast-like format. It evaluates along three dimensions: text-based, speech-based, and audio-based, and introduces additional metrics beyond the objective metrics traditionally used in speech synthesis and LLM evaluation, allowing dialogue generation models to be assessed using multiple objective metrics. For subjective evaluation, it also proposes a human evaluation procedure with filtering methods and a structured questionnaire to improve the reliability of evaluating long-form dialogue. Using the proposed evaluation framework, the paper evaluates several dialogue generation models, including PodAgent, GPT4, NotebookLM, and Mooncast, as well as real human dialogue, and presents interpretations based on the results of this evaluation.

**Strengths:**

- The research goal is well motivated. Establishing a unified evaluation framework for long-form dialogue generation seems important.
- Presents a reliable subjective evaluation procedure and analyzes its relationship to each model’s performance.
- Provides consistent objective metrics across multiple dialogue generation models, enabling results to be analyzed in a consistent way.
- Proposes a method for subjective evaluation of long-form dialogue generation by stitching together 1-minute FIRST / MIDDLE / FINAL segments, allowing assessment without length bias.
- Open-sourced evaluation framework

**Weaknesses:**

- Text-based and speech-based evaluations are already well established and not particularly original. Because podcasts are fundamentally about spoken communication, at this stage the use of audio-based evaluation for dialogue generation feels less motivated to me. What seems more important is proposing evaluation methods that capture how well the generated podcast actually communicates, including how natural, engaging, and listenable it is for a human listener.
- For long-form dialogue generation, it would have been better to see more reliable and strongly correlated objective metrics at the speech level that can actually reflect what we care about. The current speech-related objective metrics are largely based on existing, standard approaches.
- It is unclear whether other dialogue generation models will actually adopt this framework to evaluate themselves. The text-based evaluation is basically an evaluation of the LLM that produced the base script, so for models that generate dialogue conditioned on a given LLM script, the speech-based evaluation feels more important. However, the speech-related objective metrics mostly feel like pre-existing methods. If the authors want this evaluation framework to be followed, it would help to present a stronger motivation for why this specific setup should be used.

**Questions:**

- “Regarding proposed SPTD”: SPTD mostly just reflects that the speakers don’t sound identical, and as long as voices aren’t confusingly similar, pushing that gap larger doesn’t automatically mean the dialogue is clearer. So using it as direct evidence of “better clarity” feels overstated. In addition, SPTD does not capture within-speaker behavior. Measuring within-speaker segment-level similarity over time could help evaluate whether a dialogue system can generate expressive but identity-consistent voices across different emotional or conversational states, which is important for evaluating naturalness in dialogue generation.
- When evaluating a dialogue generation model, shouldn’t we also evaluate what the model actually produces, for example by running ASR on the generated audio and letting an LLM judge those transcripts, instead of only evaluating the base script?

---

> ### Author Response · Authors · 2025-11-20
> **Response to Reviewer 6Cwy**
>
> ### Dear Reviewer,
>
> Thank you for your valuable time invested in evaluation of our manuscript and providing detailed feedback. We are encouraged by your recognition of our contributions, including that *"the research goal is well motivated"*, *"establishing a unified evaluation framework for long-form dialogue generation"*, *"reliable subjective evaluation procedure"*, *"consistent objective metrics across multiple dialogue generation models"* and the practical value of *“open-sourced evaluation framework”*.
>
> We equally appreciate your constructive questions, which represent invaluable opportunities to enhance the clarity and impact of our work. Here we would like to address these points and try to avoid any misunderstanding:
>
> ---
>
> ### Q1: Discussion on reusing existing metrics
>
> **A1:** Open-ended long-form audio evaluation is a highly challenging task, although we leveraged some existing metrics in our evaluation framework, the contribution of this work goes far beyond that, for example:
>
> 1. **Conceptual decomposition of a complex task.** A main innovation of our approach is to decompose the task into multiple interpretable dimensions. It is unrealistic to evaluate long‑form audio with a single metric. Such audio involves many different aspects — linguistic informativeness, acoustic naturalness, speech stability, dialogue dynamics, listener perception and so on. Therefore, we decompose this complex task into two fundamental perspectives, **“content”** and **“form”**, which are implemented through our systematic **text‑speech‑audio** three‑dimensional evaluation framework.
>
> 2. Since it is an **open‑ended generation task**, there is no single ground‑truth reference available.  We construct a **real-world podcast reference dataset** spanning a wide range of podcast categories and topics, serving as a reference for *human-level creative quality*. This dataset provides essential ground truth for evaluation and enables meaningful comparison between AI-generated and human-created content.
>
> 3. We carefully design a comprehensive **subjective evaluation framework** to ensure **reliable and interpretable** human assessments.The framework includes:
>
>    a. **Naturalness assessment** for dialogue‑level speech segments.
>    b. **Multi‑dimensional feedback** on overall long‑form audio, covering engagement, information‑delivery effectiveness, willingness to listen to the full episode, and other key aspects.
>    c. **Strategic selection of evaluation methods**:
>       – For *dialogue naturalness*, we adopt a *MUSHRA‑based testing framework* and design reference, high‑quality, and low‑quality samples for pairwise comparison, which improves the reliability and confidence of subjective judgments.
>       – For *overall audio evaluation*, we use a *questionnaire‑based assessment* that balances coverage of essential dimensions and practical time constraints.
>    d. *Unified data formatting* to accommodate variable‑length, open‑ended audio generations.
>    e. *Attention‑check mechanisms* to verify data validity and ensure the robustness of human ratings.
>
> 4. For those established metrics, we bridge the gap between "having a metric" and **"making it practically applicable"** for long-form podcast evaluation. For example, Speaker Similarity (SIM) is commonly used in zero-shot TTS evaluation, but standard applications simply compute similarity between reference and target audio. In the podcast evaluation context, to make the evaluation framework applicable for any podcast generation system, we deal with the final complete podcast audio rather than the intermediate speech segments. Therefore, we design a process pipeline: background sound removal → speaker boundary detection → segment extraction → same-speaker clustering → SIM computation. During the process, we further optimize the effectiveness by filtering short segments that providing insufficient suprasegmental information for accurate speaker representation.
>
> In summary, our work goes beyond reusing existing metrics — it establishes a comprehensive **conceptual** and **practical** framework for **open‑ended long‑form audio evaluation**. Moreover, all components — including data processing workflows, objective metric implementations, and the web interface for subjective evaluations — have been **fully open‑sourced**. While some implementation details may not be fully presented in the paper due to space limitations, we hope the above discussion helps to **clarify our contributions** and address your concerns.

---

> > ### Author Response · Authors · 2025-11-20
> > **Response to Reviewer 6Cwy**
> >
> > ### Q2: The motivation of audio-based evaluation
> >
> > **A2:** Yes, speech is undoubtedly the core component of a podcast. However, if we focus solely on speech evaluation, the task becomes more akin to assessing a text‑to‑speech (TTS) system rather than evaluating a podcast itself. In this work, our goal is to establish a **comprehensive podcast evaluation framework** from a **real user’s perspective**.
> >
> > From a listener’s standpoint, the primary motivation for choosing a podcast is to **receive information** — not only through *what* is said but also from *how* it is presented. Therefore, both **content** and **form** play essential roles in real‑world podcast perception. Based on this understanding, we decompose the evaluation into **three modalities: text, speech, and audio**.
> >
> > It is important to note that **speech‑based** and **audio‑based** evaluations are **complementary rather than orthogonal** in long‑form audio assessment. The audio‑based evaluation is not merely about non‑speech elements (e.g., background music); instead, it reflects the **holistic auditory experience** that influences user engagement. For instance, even when a podcast consists solely of spoken content without background sound, an audio‑level evaluation remains essential.
> >
> > Specifically, the **speech‑based evaluation** focuses on the **naturalness and robustness** of conversational segments, while the **audio‑based evaluation** assesses the **overall listening experience** — including perceived audio quality, effectiveness of information delivery, listener engagement, the willingness to continue listening to the entire long‑form podcast and so on. These aspects are integral to capturing the true characteristics of long‑form podcast generation.
> >
> > ---
> >
> > ### Q3: Discussion on objective metrics at the speech level
> >
> > **A3:** Yes, designing objective metrics that can directly reflect human perception of speech—such as naturalness—is undoubtedly valuable. There are studies specifically devoted to training models to **approximate human judgment scores**, aiming to build learnable perceptual metrics. However, at present, most reliable and interpretable objective measures still rely on having a **standard reference** as guidance. For **naturalness**, especially at the **dialogue level**, no existing objective method can yet fully replace human evaluation.
> >
> > We envision integrating such advanced perceptual metrics into our framework in the future. However, in this work, our primary goal is to establish a **comprehensive and practically applicable evaluation framework** that aligns closely with **real‑world user needs**, while ensuring that its evaluation results are **reliable and usable in practice**. To achieve this, we have devoted significant effort to the **design of subjective evaluation protocols** as we mentioned in Answer **A1-(3)**.
> >
> > In our framework, **objective and subjective evaluations** are **complementary and mutually reinforcing**, rather than substitutive. For example, metrics such as *WER* can quantitatively assess model robustness. Only when a system reaches sufficient stability does subjective evaluation of *naturalness* become truly meaningful. This progressive relationship not only enhances evaluation reliability but also **reduces the human evaluation cost**.
> >
> > Importantly, evaluating **open‑ended long‑form audio** is inherently a **challenging task**, even from a human listener’s perspective. This further underscores the necessity of establishing an **effective, user‑centric evaluation framework** that can enable the collection of **richer and more reliable human‑preference data**, ultimately guide the development of future **automated naturalness evaluation methods**.
> >
> > ---
> >
> > ### Q4: Applicable to other dialogue generation models
> >
> > **A4:** Absolutely. This work is **designed to be generally applicable** to any *podcast‑like long‑form audio generation system*. More precisely, our framework is developed to evaluate **the generated long audio itself**, regardless of what type of model produces it. The underlying generative model can vary widely — our framework remains applicable as long as the output is a long-form, dialogue‑style audio segment.
> >
> > All evaluation metrics in our work are **derived from real application scenarios** and built upon **real users’ expectations** of what constitutes a high‑quality podcast experience. While dialogue speech generation models are currently the most common type of systems used to produce such long-form content, our evaluation framework naturally encompasses them as a key category.
> >
> > As summarized in **Table 1** in the paper, we have already tested multiple representative systems, including NotebookLM, Dia, Muyan‑TTS, MoonCast, MOSS‑TTSD, and PodAgent — among which Dia, MoonCast, and MOSS‑TTSD are dialogue speech generation models.

---

> > > ### Author Response · Authors · 2025-11-20
> > > **Response to Reviewer 6Cwy**
> > >
> > > ### Q5: Discussion on SPTD
> > >
> > > **A5:** We agree that a higher SPTD does not always imply better clarity. However, an excessively low SPTD indicates that the timbres of different speakers are overly similar, which can lead to perceptual confusion and reduced dialogue distinguishability. Therefore, SPTD can be used as an indicator of whether the generated speaker timbres are sufficiently distinguishable to avoid confusing overlaps in multi‑speaker long‑form content. In real‑world podcast‑like scenarios, such perceptual separability is a fundamental prerequisite for listeners to follow a conversation naturally.
> > >
> > > ---
> > >
> > > ### Q6: Within-speaker segment-level similarity
> > >
> > > **A6:** That’s a very good suggestion! Besides SPTD, we also provided the *SIM* metric to measure the similarity between the generated voice and its corresponding reference. However, the *within‑speaker segment‑level similarity* (denoted as **WSIM**) is a different metric that reflects whether *the dialogue system can generate expressive yet identity‑consistent voices across different emotional or conversational states* as you said. We sincerely thank you for this valuable feedback. We have added the WSIM results in **Appendix A.7** in the revised PDF for reference, and we will integrate the implementation into our open‑source repository later.
> > >
> > > ---
> > >
> > > ### Q7: Discussion on evaluating what the model actually produces, e.g. ASR+LLM
> > >
> > > **A7:** That’s exactly what we do! As also described in **Q4**, our framework evaluates what the model actually produces. We understand your point as mainly asking whether we should directly evaluate the **ASR results** of the generated audio instead of the **base scripts** for text-based evaluation.
> > >
> > > The difference between the base script and the ASR output essentially reflects the **stability and robustness of the speech generation model**—and this is exactly what our *WER* metric captures in the **speech-based evaluation** (Section 5). In contrast, the **text-based evaluation** focuses on assessing the **informativeness of the intended message** itself. Therefore, we evaluate textual quality using the base scripts, and measure the model’s speech stability with **WER**, which compares the ASR transcriptions against the base scripts.
> > >
> > > This **decoupled evaluation design** makes the purpose of each metric more explicit, ensures each evaluation targets a well‑defined aspect of system behavior, and yields clearer, more interpretable analysis results. We hope this explanation addresses your concern and clarifies our evaluation rationale.
> > >
> > > ---
> > >
> > > Thank you for your insightful feedback and recognition. Your suggestions have been invaluable in helping us refine our presentation and strengthen our analysis. We hope that our responses have addressed your concerns and clearly demonstrated the potential impact of our work. We would greatly appreciate it if these clarifications could be considered in your final evaluation, and we sincerely welcome any further questions or suggestions you may have.

---

> > > > ### Author Response · Authors · 2025-11-28
> > > >
> > > > **Dear Reviewer:**
> > > >
> > > > Thank you very much for your valuable comments. We have carefully addressed each of your questions. As the timeline is somewhat tight, please feel free to let us know if you have any further questions or need additional information—we would be happy to assist at any time. We sincerely look forward to your feedback and would be honored if you found our responses merit positive consideration. Thank you again for your time and effort.

---

### Meta-Review · Area_Chair_exeg · 2025-12-30

**Summary:**

- The paper is mostly reusing existing objective metrics [6Cwy,WWEy]
- Necessity of audio‑based evaluation [6Cwy]
- Suitability of objective metrics for naturalness [6Cwy]
- Applicability to other dialogue generation models [6Cwy]
- Explaining the validity of SPTD [6Cwy,WWEy]
- Objective/ subjective correspondence in general [d89n]
- Within‑speaker consistency addressed by introducing WSIM [6Cwy]
- Transferability to other long-form audio types [WWEy]
- Analysis audio-language models' evaluation ability [WWEy]
- Reliability of subjective tests [WWEy]
- Clarifying misunderstanding regarding potential evaluation bias toward PodAgent [d89n]
- Dataset curation process [13hU]
- Novelty [6Cwy, d89n]

**Reviewer Concerns:**

Addressed:
- concerns about bias and suitability of the general approach
- concerns about dataset construction, generalizability, transferability

Unaddressed (from d89n, and shared by 6Cwy and the meta-reviewer):
- Limited methodological novelty: the framework largely aggregates existing metrics (WER, SIM, DNSMOS) rather than proposing novel evaluation methodologies tailored for the specific challenges of long-form audio (e.g., narrative coherence, long-term prosody).
- Predictive validity: the fact that a system can score perfectly on objective metrics (like SPTD/SMR) but fail subjectively limits the framework's utility for rapid, automated model iteration. The correlation of r=0.53 for SPTD, while positive, suggests it is a somewhat weak signal for quality.

The paper presents a solid engineering effort and a valuable resource, but the scientific novelty and the predictive validity of the proposed objective metrics are marginally below the bar for a top-tier conference like ICLR. It would be a better fit for a domain conference, rather than ICLR.

**Reviewer Scores:**

6Cwy: 4 -> 4 (likely)
WWEy: 6 -> 6+
d89n: 2 -> 4 (commented in text)
13hU: 8

---

### Decision · Program_Chairs · 2026-01-26

Reject